



# Can the assimilation of water isotopologue observation improve the quality of tropical diabatic heating and precipitation?

Farahnaz Khosrawi[1], Kinya Toride[2], Kei Yoshimura[2], Christopher J. Diekmann[1], Benjamin Ertl[1,3], Frank Hase[1], and Matthias Schneider[1]

[1]Institute of Meteorology and Climate Research (IMK), Karlsruhe Institute of Technology, Karlsruhe, Germany
[2]Institute of Industrial Science, University of Tokyo, Chiba, Japan
[3]Steinbuch Centre for Computing (SCC), Karlsruhe Institute of Technology, Karlsruhe, Germany
**Correspondence:** Farahnaz Khosrawi (farahnaz.khosrawi@kit.edu)

**Abstract.** The strong coupling between atmospheric circulation, moisture pathways and atmospheric diabatic heating is responsible for most climate feedback mechanisms and controls the evolution of severe weather events. However, diabatic heating rates obtained from current meteorological reanalysis show significant inconsistencies. Here, we theoretically assess with an Observation System Simulation Experiment (OSSE) the potential of the MUlti-platform remote Sensing of Isotopo-

logues for investigating the Cycle of Atmospheric water (MUSICA) Infrared Atmospheric Sounding interferometer (IASI) mid-tropospheric water isotopologue data for constraining uncertainties in meteorological analysis fields. For this purpose, we use the Isotope-incorporated General Spectral Model (IsoGSM) together with a Local Ensemble Transform Kalman Filter (LETKF) and assimilate synthetic MUSICA IASI isotopologue observations. We perform two experiments consisting each of two ensemble simulation runs, one ensemble simulation where we assimilate conventional observations (temperature, humidity

and wind profiles obtained from radiosonde and satellite data) and a second one where we assimilate additionally to the conventional observations the synthetic IASI isotopologue data. In the second experiment, we perform one ensemble simulation where only synthetic IASI isotopologue data are assimilated and another one where no observational data at all are assimilated. The first experiment serves to assess the impact of the IASI isotopologue data additional to the conventional observations and the second one to assess the direct impact of the IASI isotopologue data on the meteorological variables, especially on the heat-

ing rates and vertical velocity. The assessment is performed for the tropics in the latitude range from 10°S to 10°N. When the synthetic isotopologue data are additionally assimilated, we derive in both experiments lower Root-Mean-Square Deviations (RMSDs) and improved skills with respect to meteorological variables (improvement by about 8-13%). However, heating rates and vertical motion can only be improved throughout the troposphere when additionally to IASI $\delta$D conventional observations are assimilated. When only IASI $\delta$D is assimilated the improvement in vertical velocity and heating rate is minor (up to a few

percent) and restricted to the mid-troposphere. Nevertheless, these assimilation experiments indicate that IASI isotopologue observations have the potential to reduce the uncertainties of diabatic heating rates and meteorological variables in the tropics and in consequence offer potential for improving meteorological analysis, weather forecasts and climate predictions in the tropical regions.



## 1 Introduction

In the past 40 years, medium-range weather forecasts have undergone significant improvements (Bauer et al., 2015). The improvement in forecast skill is mainly due to improved data assimilation techniques, model dynamics and physics, spatial and temporal model resolution, representation of uncertainties, together with improved observing systems whose measurements can be used for data assimilation. Nevertheless, a correct initialisation and the model used for the creation of the initial conditions are crucial for the quality of the weather forecast (Magnusson et al., 2019).

Meteorological analysis and reanalysis are best guesses of the true state of the atmosphere. Therefore, these are of great importance for both, the initialisation of weather forecasts and for process analysis and detection and attribution of changes in the climate system (Wright and Fueglistaler, 2013). Diabatic heating is the major driving force of atmospheric circulation on weather and climate time scales. However, diabatic heating rates obtained from current meteorological reanalysis show significant inconsistencies (e.g. Chan and Nigam, 2009; Ling and Zhang, 2013; Wright and Fueglistaler, 2013). This jeopardises the accuracy of both climate predictions and numerical weather prediction. This is mainly indebted to the fact that diabatic heating rates cannot be observed directly. Usually, diabatic heating is diagnosed from the atmospheric circulation (wind and temperature) through the thermodynamic energy equation (Peixoto and Oort, 1992; Chan and Nigam, 2009).

Water isotopologue observations (e.g. $H_2O$ and HDO) assimilated into meteorological reanalysis can make an invaluable contribution since the isotopologue composition depends on the history of phase transition. Stable water isotopologue ratios are sensitive to the phase changes during atmospheric circulation. Therefore, water isotopologues are useful tracers for investigating atmospheric processes, such as large-scale transport (e.g. Yoshimura et al., 2003; Lee et al., 2017; Risi et al., 2012; Dee et al., 2018) and cloud-related processes (e.g. Webster and Heymsfield, 2003; Worden et al., 2007). The relation between atmospheric processes and isotopic information in water vapour and precipitation has therefore been studied intensively (e.g. Yoshimura et al., 2004; Schneider et al., 2016; González et al., 2016; Lacour et al., 2017; Risi et al., 2019). Further, isotopologue observations can provide information that is closely linked to diabatic heating processes (e.g. Lacour et al., 2018; Risi et al., 2020).

First attempts for testing the impact of assimilating water isotopologues were done by Yoshimura et al. (2014) and Toride et al. (2021). Yoshimura et al. (2014) developed a new data assimilation system using a Local Ensemble Transform Kalman Filter (LETKF) and the Isotope-incorporated Global Spectral Model (IsoGSM). They then applied this assimilation system to an Observation System Simulation Experiment (OSSE) using a synthetic data set that mimicked water vapour isotope measurements from the Tropospheric Emission Spectrometer (TES), the SCanning Imaging Absorption spectroMeter for Atmospheric CHartographY (SCIAMACHY) and the Global Network of Isotopes in Precipitation (GNIP). Their results showed that not only the water isotopic fields were improved but also the meteorological fields (e.g. temperature, pressure, wind speed).

In the study by Toride et al. (2021) the same OSSE was used, but synthetic isotope data from the Infrared Atmospheric Sounding interferometer (IASI) were assimilated in addition to conventional non-isotopic observations. Their results showed that the additional assimilation of the water isotopologue information leads to a further improvement of the meteorological





fields compared to the assimilation of conventional data alone. Furthermore, the large-scale atmospheric circulation and the weather forecast could be improved.

Here, we apply the same assimilation experiment that was used in Toride et al. (2021). While Toride et al. (2021) had their
focus on the impact on large-scale circulation and weather forecasts on a global scale, we investigate here the impact of the assimilation of the IASI isotopologue data on the meteorological analysis fields in the tropics. Especially, we are interested to answer the following question: Can the assimilation of IASI water isotopologues help to improve diabatic heating rates and/or precipitation rates? We investigate here both, the benefit of IASI water isotopologue data being assimilated additional to conventional observations and the direct impact the IASI water isotopologues data have when these are assimilated alone
without considering any other observations. To directly asses the impact of the assimilation of the IASI water isotopologue data we use an additional OSSE where only IASI isotopologue information is assimilated and compare this to an ensemble simulation where no observations at all are assimilated.

The paper is structured as follows. In Sect. 2 we describe the data and method used. In Sect. 3 we a assess the performance of the assimilation experiment with IASI water isotopologues additional to the conventional observations for the tropics and for
specific longitude regions in the tropics. This is then followed by an assessment of the direct impact of the assimilation of IASI water isotopologues on the meteorological analyses, especially on diabatic heating and precipitation. In Sect. 4 we discuss and in Sect. 5 we summarize and conclude our results.

## 2  Data and Method

### 2.1  IASI observations

Isotopologues observations from the Infrared Atmospheric Sounding Interferometer (IASI) onboard the Meteorological Operation Satellites A and B (MetOp-A and MetOp-B) are used (Schneider and Hase, 2011; Schneider et al., 2016; Diekmann et al., 2021a). IASI is a nadir-looking Fourier-transform spectrometer and measures in the infrared part of the electromagnetic spectrum. The IASI instrument has its main purpose in providing observational data to support numerical weather prediction, however, due to the instruments high signal-to-noise ratio and high spectral resolution the instruments enables also atmospheric
trace gas observations as e.g. carbon monoxide (CO), methane ($CH_4$), nitric acid ($HNO_3$) (e.g. Clerbaux et al., 2009; Schneider et al., 2021).

IASI measurements are obtained with a horizontal resolution of 12 km (pixel diameter at nadir viewing geometry) with a swath width of 2200 km and 14 sun-synchronous orbits per day. We use the retrieval recipe of the MUlti-platform remote Sensing of Isotopologues for investigating the Cycle of Atmospheric water (MUSICA). The free tropospheric water vapour
isotopologue composition can be retrieved from IASI spectra measured during cloud free conditions (Schneider et al., 2016; Diekmann et al., 2021a). The ratio $R$ between the isotopologues HDO and $H_2O$ is given in the $\delta$ notation and $\delta$D is then calculated with reference to the Vienna Standard Mean Ocean Water $R_s$ ($\delta$D = $1000 \times (R/R_s - 1)$ in ‰, with $R_s$ =3.1152 × $10^{-4}$). IASI $\delta$D and $H_2O$ pair distributions are provided twice a day (each about 300 000 points) on a quasi-global scale with 11 vertical layers from 1.3 to 8.0 km.





Only IASI data of high quality is used, e.g measurements at 4.2 km, the altitude where IASI $\delta$D has the highest sensitivity and filtering of the data by applying several quality filters (e.g. measurements response, cloud-free scenes). The data are spatially resampled to the IsoGSM grid at one vertical sigma level corresponding to 4.2 km (Toride et al., 2021).

## 2.2   IsoGSM model and data assimilation

For our assimilation experiments we use the isotope-incorporated Global Spectral Model (IsoGSM). This model is based on
the Scripps Experimental Climate Predictions Center's (ECPC) Global Spectral Model (GSM) that has been used by NCEP to perform operational analyses and medium-range forecasts (Kanamitsu et al., 2002). Gaseous forms of stable water isotopes (HDO and $H_2^{18}O$) are incorporated as prognostic variables in addition to water vapour into GSM (Yoshimura et al., 2008). Simulations with IsoGSM have been used together and also evaluated with both, ground-based (Schneider et al., 2010; Uemura et al., 2008) and space-borne observations (Frankenberg et al., 2009; Yoshimura et al., 2011) of water isotopologues.

Here, we use IsoGSM ensemble simulations performed with a T62 horizontal resolution (1.9°$\times$ 2°, $\sim$200$\times$200 km) and 28 vertical sigma levels from the surface up to $\sim$2.5 hPa. The sea surface and sea ice temperature distribution from the National Centers of Environmental Prediction/Department of Energy Reanalysis 2 (NCEP-DOE, Reanalysis 2) have been used as lower boundary condition. The data assimilation is performed with a Local Ensemble Kalman Transform Filter (LETKF, Hunt et al. (2004, 2007)) which is a parallel-efficient update of the traditional Ensemble Transform Kalman Filter (ETKF,
Bishop et al. (2001)). For the data assimilation a relaxation-to-prior spread (RTPS) method (Whitaker and Hamill, 2012) is used with a relaxation parameter of 0.4 to maintain an appropriate ensemble spread and to avoid filter divergence. The horizontal localisation scale is set to be 500 km (influence radius of 1826 km for best assimilation performance). The used ensemble size is 96. This large ensemble size is needed to derive results of satisfying quality as was shown by Toride et al. (2021). A detailed description on the data assimilation with LETKF and IsoGSM is provided in Yoshimura et al. (2014) and
Toride et al. (2021).

## 2.3   Observation System Simulation Experiment

To investigate the potential impact of the assimilation of satellite data an Observation System Simulation Experiment (OSSE) is performed. In an OSSE, a model simulation is regarded as "truth" ("Nature run") and several data assimilation experiments with synthetic observations derived from the Nature run are conducted that aim to reproduce the Nature run as closely as possible
(Schröttle et al., 2020). The synthetic data mocks therefore the data that would be obtained if satellites or ground-based sensors were actually operated (Yoshimura et al., 2014).

For the OSSEs performed in this study the characteristics of the IASI $\delta$D observations generated by the MUSICA IASI retrieval processor are mocked. For generating the synthetic MUSICA IASI data set the spatial coverage and the observational error statistics of the real data are used. Our mocked "truth" data set has been derived from an IsoGSM simulation and is used as
reference for assessing the impact of our assimilation experiments on the meteorological variables. The synthetic observational data set is then generated by adding Gaussian noises with the actual error statistics of the IASI observations to the Nature run (Toride et al., 2021).



## 2.4 Experimental set-up

Two experiments consisting in total of four ensemble simulations (two per experiment) were performed to investigate the poten-
tial impacts of assimilating IASI water vapour isotopologues observations. Thereby, three OSSEs and one ensemble simulation
without any data assimilation were performed to assess the potential impact of the IASI $\delta$D data on the meteorological fields.
In the first OSSE synthetic conventional observations of temperature, humidity and wind profiles obtained from radiosonde
and satellite data are assimilated, in the second OSSE synthetic IASI observations are assimilated additionally to the conven-
tional observations and in the third OSSE synthetic IASI data are assimilated alone. The former two OSSEs serve to assess
the additional benefit one gets if to the conventional assimilation procedure used in numerical weather prediction IASI data
would be assimilated. The third one serves to investigate what impact the assimilation of the IASI $\delta$D data alone has on the
meteorological fields, e.g. on the diabatic heating rates. Therefore, this OSSE is compared to an ensemble simulation without
assimilation of any observations.

     The Nature run is generated by an IsoGSM simulation over two years, covering the time period from 2015 to 2016. The
model run was started on 1 June 2015 at 00 UTC. The first year has been discarded as spin-up period to minimize the possibility
of the model's drift. The initial conditions for the 96 ensemble members were taken from the Nature run. The first initialisation
was done on 1 June 2016 at 00 UTC and then all other ensemble members were initialised with the following consecutive 6-
hour time steps. Therefore, the initial conditions can be considered as being independent from the Nature run, but representing
similar climatological conditions. The following two months, thus from 1 July 2016 to 1 September 2016 have then been used
as the experimental period and the results of our assimilation experiments are then evaluated for the latter one-month period (1
August to 31 August 2016).

     The synthetic conventional observations (radiosondes, wind profilers, aircrafts, ships, buoys, surface stations, and wind
data derived from satellites and radar) are generated based on a data set used in the NCEP operational system (known as
PREPBUFR, i.e. preprocessed and quality controlled observations of the Binary Universal Form for the Representation of me-
teorological data (BUFR), https://rda.ucar.edu/datasets/ds337.0/). PREPBUFR is a commonly used data set in data assimilation
studies (e.g. Koshin et al., 2020). The conventional observations are also spatially resampled to the IsoGSM grid.

     In the following the ensemble simulation with the assimilation of the conventional observations is called DA_prepbufr,
the one where additionally to the conventional observations IASI $\delta$D is assimilated is called DA_prepbufr_IASI, the one
with the assimilation of IASI $\delta$D alone is called DA_IASI and the ensemble simulation without any data assimilation is
called noDA. When we later in Section 3.3 compare the former two with the latter two ensemble simulations we call the first
assimilation experiment (consisting of the ensemble simulations DA_prepbufr and DA_prepbufr_IASI) PREPBUFR and the
second assimilation experiment (consisting of the ensemble simulations DA_IASI and noDA) noDAvsDA. An overview over
our assimilation experiments is given in Table 1.

     The assessment of the idealized assimilation experiment is then done by using each experiments ensemble mean, the mean
difference between each ensemble mean of the respective assimilation run and the Nature run, the root-mean-square deviation
(RMSD) between each experiments ensemble mean and the Nature run and the RMSD skill. The ensemble mean and the mean




difference between the assimilation run and the Nature run are calculated by:

$$\bar{x} = \frac{1}{N} \sum_{i=1}^{N} x_i \tag{1}$$

and

$$MD = \frac{1}{N} \sum_{i=1}^{N} (x_i - x_{n_i}) \tag{2}$$

where $x$ denotes the assessed meteorological variable (e. g. $T$, $u$, $v$) of the assimilation experiment and $x_n$ the respective meteorological variable of the Nature run. The RMSD is calculated as follows:

$$RMSD = \sqrt{\frac{1}{N} \sum_{i=1}^{N} (x_i - x_{n_i})^2} \tag{3}$$

The skill (in %) is calculated by:

$$Skill = \frac{RMSD_{CTRL} - RMSD}{RMSD_{CTRL}} \cdot 100 \tag{4}$$

where CTRL denotes the assimilation with the conventional observations (DA_prepbufr in case of the PREPBUFR experiment) and the ensemble simulation without any data assimilation (noDA in case of the noDAvsDA experiment), respectively. RMSD and skill are typical measures for the quality of a simulation that are commonly used in Numerical Weather Prediction (NWP, e. g. Bauer et al. (2015))

## 2.5  IsoGSM output and derived parameters

For the assessment of the assimilation experiments we use the IsoGSM output of the following parameters: temperature ($T$), zonal ($u$) and meridional wind ($v$), vertical velocity ($\omega$), specific humidity ($q$) and precipitation. The water isotopologues $\delta$D and $\delta^{18}$O are derived from converting the model output of HDO and H$_2^{18}$O from mixing ratios to the delta notation in per mille. The apparent heat flux of the large scale motion system $Q_1$ and the apparent moisture sink $Q_2$ which is due to the net

**Table 1.** Specification of the experiments used in this study. Checkmarks indicate the variables assimilated in each experiment.

| experiment | assimilation run | ensemble size | conventional observations | IASI $\delta$D ($\sigma$=0.568) | localization (km) | Inflation | RTPS |
|---|---|---|---|---|---|---|---|
| PREPBUFR | DA_prepbufr | 96 | ✓ | | 500 | 1.05 | 0.4 |
| | DA_prepbufr_IASI | 96 | ✓ | ✓ | 500 | 1.05 | 0.4 |
| noDAvsDA | noDA | 96 | - | - | - | - | - |
| | DA_IASI | 96 | - | ✓ | 500 | 1.05 | 0.4 |





condensation and vertical divergence of the vertical eddy transport of moisture are calculated based on the equations given in
Yanai et al. (1973).

$$Q_1 = \frac{\partial s}{\partial t} + \boldsymbol{V} \cdot \nabla s + s\frac{\partial s}{\partial p} \tag{5}$$

$$Q_2 = -L\left(\frac{\partial q}{\partial t} + \boldsymbol{V} \cdot \nabla q + \omega\frac{\partial q}{\partial p}\right) \tag{6}$$

where $s$ is the dry static energy, $\omega$ the vertical velocity, $L$ is the latent heat of net condensation, $q$ the specific humidity, $\boldsymbol{V}$ the
horizontal wind vector and $p$ the pressure.

## 3  Results

### 3.1  Assessment of the performance in the tropics

The assessment is performed for the tropics in the latitude range from 10°S to 10°N and for the one month period of August
2016. In the following this experiment is called PREPBUFR and the assimilation run with the conventional observations is
called DA_prepbufr and the one with the additional assimilation of IASI $\delta$D data is called DA_prepbufr_IASI. Figure 1 (left)
shows the spatial and temporal averaged vertical profiles of the ensemble mean for $\delta$D, moisture sink ($Q_2$) and vertical velocity
($\omega$) for the tropics (one month average over all longitudes). The spatially and temporally averaged ensemble mean profiles
reflect the characteristics of the tropics, mainly upwelling ($\omega$), drying above 800 hPa and moistening below and heating ($Q_2$)
in most parts of the troposphere. In the spatial and temporally averaged ensemble mean profiles differences between the Nature
and the assimilation runs are quite low and become only visible when the mean difference between the assimilation run and
the Nature run is considered (Fig. 1 right). Generally, for the assimilation with additionally IASI $\delta$D (DA_prepbufr_IASI) the
mean differences are lower than for the assimilation run with conventional observations only (DA_prepbufr).

Considering the corresponding RMSD and skill, which are shown in Fig. 2 for $\delta$D, $Q_2$ and $\omega$, we find a clear decrease in the
RMSD for the assimilation run where IASI $\delta$D is assimilated (DA_prepbufr_IASI). A decrease in the RMSD and improvement
in skill is also found for all other parameters (Fig. S2 and S3). The highest decrease in the RMSD and highest skill is found for
all parameters at ∼500-600 hPa, corresponding to the approximate altitude level where the IASI data has been assimilated. The
improvement in the skill for $\delta$D is about 6 % at the lowest altitudes and increases to almost 40 % at 600 hPa and then decreases
to 6 % at 300 hPa and remains at this value up to 100 hPa. For $Q_2$ the improvement in skill ranges between 8-10 % up to
400 hPa and decreases then to 5 %. Although the improvement in skill is mostly decreasing with altitude, this comparisons
show that at all altitudes in the troposphere the assimilation of the IASI data additionally to the conventional observations leads
to an improvement in the RMSD of the analysis variables. That this holds not only for the tropical mean profile, but also for all
longitudes in the tropics can be seen from the cross sections.

Figure 3 shows the cross sections of the RMSD for the DA_prepbufr and the DA_prepbufr_IASI assimilation runs for $\delta$D
and $Q_2$ and the absolute difference of the RMSD of the two runs and the skill. There are certain areas where high RMSDs

**Figure 1.** Spatial and temporal averaged vertical profiles of the ensemble mean (left) and the mean difference of the ensemble mean from the Nature (right): $\delta$D (top), moisture sink $Q_2$ (middle), and vertical velocity $\omega$ (bottom) for the tropics ($10°$S to $10°$N) for the Nature (grey), the experiment with assimilation of the conventional observations − DA_prepbufr (blue) and the experiment with assimilation of the mocked IASI $\delta$D data additional to the conventional observations − DA_prepbufr_IASI (red).

**Figure 2.** Spatial and temporal averaged vertical profiles of the RMSD (left) for the experiment with assimilation of the conventional observations − DA_prepbufr (blue) and the experiment with assimilation of the mocked IASI $\delta$D data additional to the conventional observations − DA_prepbufr_IASI (red) and the improvement in skill when additionally to the conventional observations the mocked IASI $\delta$D is assimilated (right): $\delta$D (top), moisture sink $Q_2$ (middle) and $\omega$ (bottom) for the tropics (10°S to 10°N).



**Figure 3.** Cross sections of the RMSD for $\delta$D (left) and $Q_2$ (right) from the assimilation experiment with the conventional observations alone (DA_prepbufr, top row), the one with assimilation of the mocked IASI data additionally to the conventional observations (DA_prepbufr_IASI, second row), the absolute difference between these two (third row) and the skill (bottom row) for the tropics (10°S to 10°N).

are found in $\delta$D and $Q_2$, but the RMSD in these areas is significantly reduced in the DA_prepbufr_IASI assimilation run. This is also clearly reflected in the absolute difference of the RMSD (with positive values showing an improvement) and also the skill for both parameters is significantly improved. For $\delta$D the highest improvement in the skill is found at around 500 hPa which is approximately the altitude where the IASI data has been assimilated (while however the highest RMSD in $\delta$D is

found in the upper troposphere at around 200-300hPa). For $Q_2$ the highest RMSD is found at the lowest atmospheric layers (1000-700 hPa) and the skill is significantly improved for DA_prepbufr_IASI throughout the troposphere (up to 20 %).





Figure 4 shows the cross sections for $Q_1$, $Q_2$ and $\omega$ for the tropics (monthly mean for August 2016). The longitudinal distribution is strongly tied to the equatorial Walker circulation which consists of several east–west circulation cells spanning different longitudinal sectors along the Equator; whereby having it's major cell above the tropical Pacific (Peixoto and Oort,

1992). Regions of diabatic heating (Fig. 4) are located in the convective regions over Asia (around 120°E), South America (around 60°W) and western and central Africa (near 20°E). Conversely, the absence of convection over the eastern Pacific leads to strong subsidence (Wright and Fueglistaler, 2013). When comparing the longitudinal regions where the high RMSD in δD is found with the vertical velocity and $Q_1$ (see Fig. 4 top panel) we find that these regions of high RMSD coincide with regions where strong upward/downward motion and diabatic heating/cooling is dominant. For $Q_2$ these regions coincide with

the regions where upward motion and heating is dominant.

The Comparison of the regions where the high RMSD is found in the cross sections (Fig. 3) with the underlying map of the monthly mean distribution of $Q_2$ and precipitation (Fig. 5) reveals the regions where the high RMSD is found in δD and $Q_2$ geographically. While the high RMSD in $Q_2$ is found over America, Asia and the Pacific, the high RMSD in δD is found over America and the Pacific. Based on this we select three regions in the tropics which will be analysed in the following

in more detail. We selected the regions over land (Asia, America, Africa) since we are also interested in the performance of the assimilation experiments with respect to precipitation and in the regions over land also the highest precipitation is found (Fig. 5). Further, these regions are characterized by different strengths of downward and upward motion which will be of interest for the assessment of the performance of our assimilation experiments in these regions. For example, parts of the Pacific Walker circulation (the western (90 to 160°E) and central (165°E to 175°W) as defined in Dee et al. (2018)) are located

in the here selected Asian region.

## 3.2 Assessment of the performance by regions

We assess the performance of the assimilation experiments in specific tropical regions and selected therefore the three regions Asia (60°E to 180°E), America (120°W to 30°W) and Africa (30°W to 60°E). Figure 4 shows the cross sections for $Q_1$, $Q_2$ and $\omega$ for the tropics and for the three selected regions separately these are shown in Fig. S4 in the supplement.

The Asian region is characterised by strong heating and upward motion. In the lowest layers (below the 800 hPa level) $Q_2$ shows a cooling (moistening) and heating (drying) above. In the Asian region the heating and upward motion are the highest of the three regions considered. The American region is characterised by both strong upward motion and heating as well as some parts of downward motion and cooling, thus showing some intermediate or balanced characteristic with alternating upward/downward motion and vertical velocities that stay around zero within the troposphere. The African region in contrast to

the other two regions is characterised by mostly downward motion and cooling and only a small area with heating and upward motion. In the African region the highest cooling (moistening) in $Q_2$ is found in the lowest layers (below the 800 hPa level).

The different characteristics of the regions are also reflected in the ensemble mean vertical profiles averaged over the respective regions and over the month August 2016. Figure 6 shows the averaged ensemble mean profiles for $Q_1$, $Q_2$ and vertical velocity. The Asian region is characterised by strong heating and upward motion. The profiles of $Q_1$ and $Q_2$ are quite similar

showing large positive values throughout the troposphere. Solely, in the lowest layers (below the 800 hPa level) $Q_2$ is negative



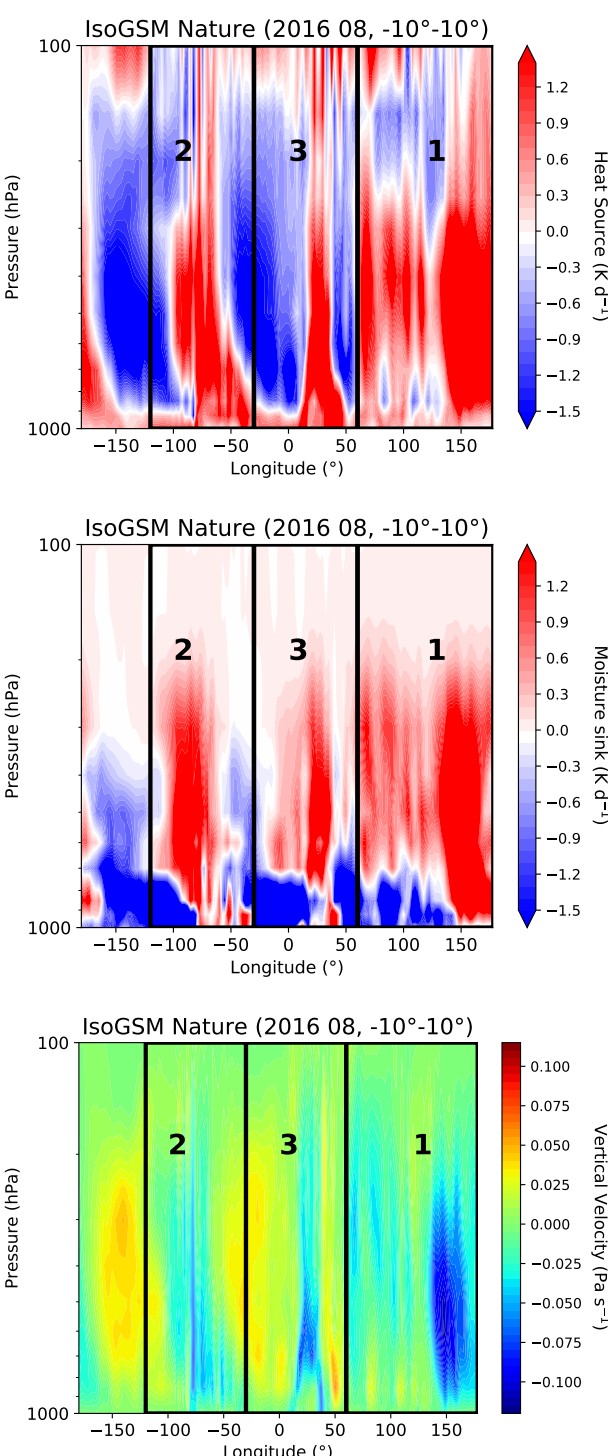

**Figure 4.** Cross sections for heat source ($Q_1$), moisture sink ($Q_2$) and vertical velocity ($\omega$) derived from the Nature run for the tropics (from top to bottom) for August 2016 ($10°$S to $10°$N). The black boxes show the selected regions in the tropics that will be considered in the further analyses: (1) Asia, (2) America, (3) Africa.

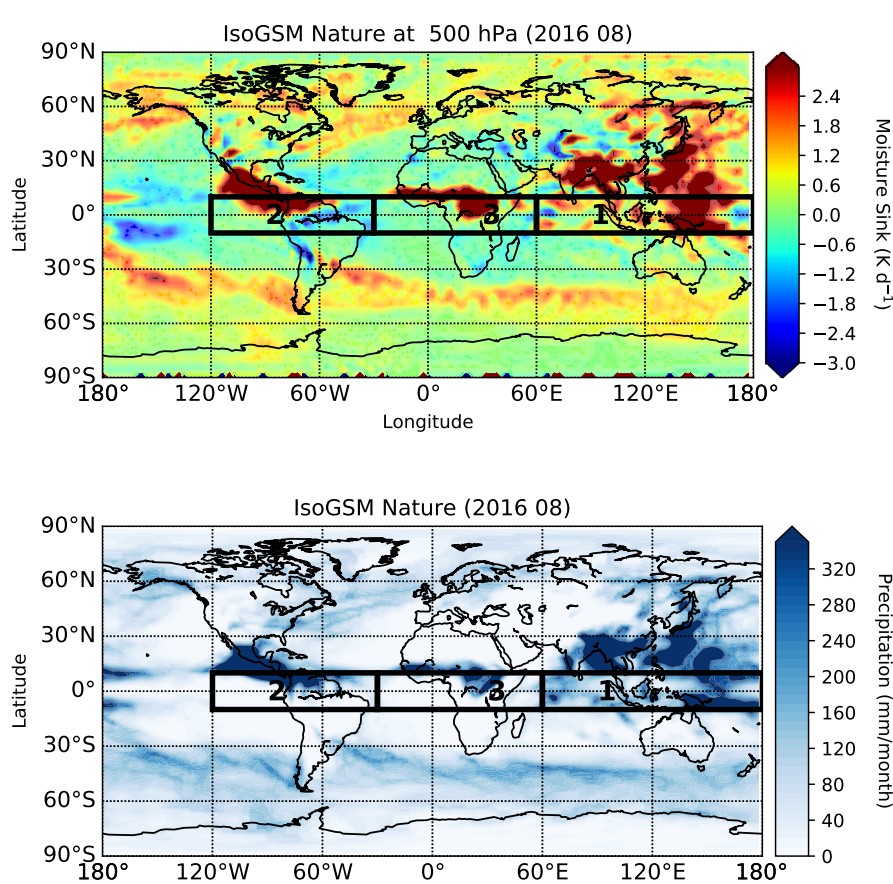

**Figure 5.** Monthly mean distribution of moisture sink ($Q_2$) and precipitation (bottom) from the Nature run at the 500 hPa level. The specific regions in the tropics that are considered in the further analyses are overlaid as black boxes: (1) Asia, (2) America, (3) Africa.

indicating moistening due to evaporation. The quite similar shape of $Q_1$ and $Q_2$ but vertically shifted peaks indicate the occurrence of deep cumulus convection within the Asian monsoon (Yanai and Tomita, 1998). In the Asian region, the heating and upward motion are the highest of the three regions considered.

The American region is characterised by both strong upward motion and heating in the lowest layers as well as downward motion and radiative cooling in the mid to upper troposphere. Thus, showing some intermediate or balanced characteristic. The heating ($Q_1$) in the lowest layers can be explained by the vertical convergence of sensible heat flux from the surface. The moistening in the lowest layers ($Q_2$) is due to evaporation (Yanai and Tomita, 1998).

    The African region in contrast to the other two regions is characterised by mostly downward motion and cooling throughout the troposphere. Solely, in the lowest layers (below 800 hPa) positive values are found in $Q_1$ indicating sensible heating. Large

negative values are found in $Q_2$ ($-1$ to $-3$ K d$^{-1}$) in the lowest layers (below 800 hPa) indicating strong moistening due to evaporation.



**Figure 6.** Spatial and temporal averaged vertical ensemble mean profiles of the heat source ($Q_1$), moisture sink ($Q_2$) and vertical velocity ($\omega$, top to bottom) for Asia (left), America (middle) and Africa (right).

Generally, the differences between the regions considered here can be described as follows: The strongest upward motion and associated heating is found in Asia, while the highest downward motion and cooling is found in the African region. The characteristics in heating and vertical motion are in America somewhere in between showing a mixture between heating/cooling and upward/downward motion. For $Q_2$ the moistening is highest in Africa and lowest in Asia. Considering the corresponding MD, RMSD and skill (Fig. S5-Fig. S7) we find as for the entire tropics that the MD and RMSD is decreased and the skill improved and that thus the run DA_prebufr_IASI is generally closer to the Nature and thus the assimilation of IASI $\delta$D improves the diabatic heating rates and vertical motion. For example, for $Q_2$ the improvement in skill is in average throughout the troposphere about 7-9 % (7.65 % for Asia), 9.97 % for America and 5.29 % for Africa, see also Tab. S1). Similar values are derived for $\omega$ (7.61 % for Asia, 10.49 % for America, 5.27 % for Africa, Tab. S1 and Fig. 7).

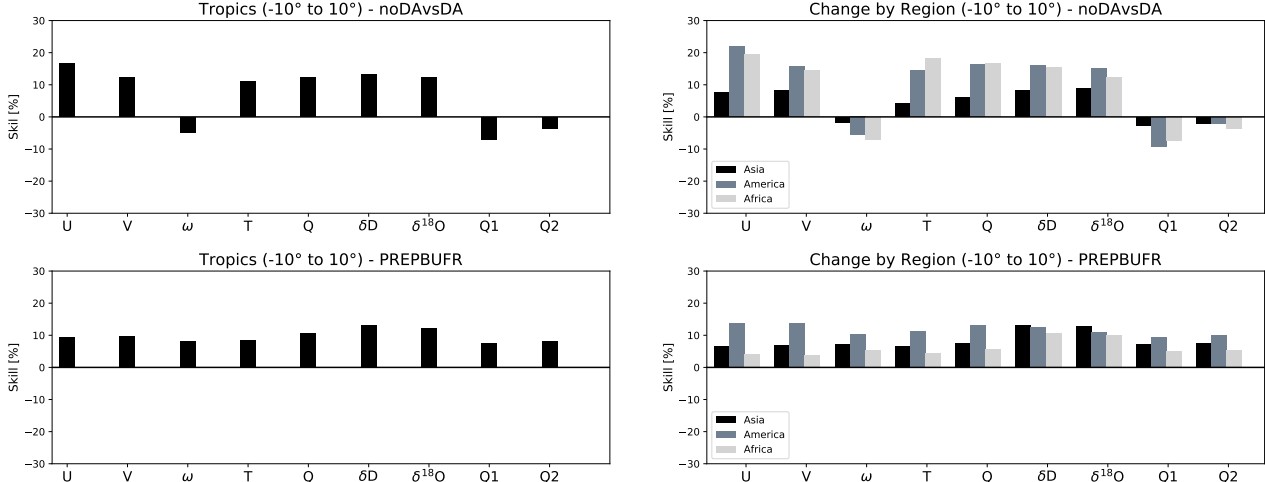

**Figure 7.** Improvement/degradation in skill in percent for each parameter in the troposphere (up to the 100 hPa level) derived from the vertical mean profiles. Top: for the simulation runs with assimilation of the mocked IASI data compared to the simulation without any data assimilation (noDAvsDA experiment). Bottom: for the simulation with assimilation of the mocked IASI additional to the conventional observations compared to the simulation runs with only assimilating the conventional observations (PREPBUFR experiment). Left: tropics, right: tropics separated by regions.

Figure 7 (bottom left) shows the improvement in skill for each parameter as bar chart and thus summarises the results of this assimilation experiment (PREPBUFR) for the tropics. We find an improvement for all parameters in the tropics of about 8-13 %. Separated by regions (Fig. 7, bottom right), we also find an improvement in skill for all parameters where the highest improvement, as for the tropics, is found for Asia and Africa for the isotopologues ($\delta$D and $\delta^{18}$O). For America the highest improvement is derived for zonal ($u$) and meridional wind ($v$) and specific humidity ($Q$).

The overall range in improvement in skill for the three regions is as follows: For Asia, except for $\delta$D and $\delta^{18}$O where the improvement is about 13 %, the improvement in skill is for the other parameters about 7-8 %. For America, the improvement is in the range of 9-14 %. For Africa the improvement in skill is about 10 % for the isotopologues ($\delta$D and $\delta^{18}$O) and 4-6 % for the other parameters (see Tab. S1). Also precipitation rates can be improved (Tab. S3). We find an improvement of 8.19 % for Asia, 13.65 % for America and 5.21 % for Africa. By region, for all here considered meteorological parameters the highest improvement is therefore found for America while the lowest improvement is found for Africa.

This again shows the benefit of additionally assimilating IASI $\delta$D to the conventional observations. Irrespective if a specific region in the tropics or the tropics as a whole are considered, a improvement of about 10 % of the meteorological analyses due to the assimilation of IASI $\delta$D additional to the conventional observations can be achieved. In the following we are interested on the direct impact of IASI $\delta$D on the meteorological analysis. Especially, we are interested in answering the following question: Can also the assimilation of only $\delta$D improve the heating and precipitation rates? Therefore, we performed an additional assimilation experiment, where only IASI $\delta$D is assimilated and derive the skill in comparison to an IsoGSM





ensemble simulation without any data assimilation. The results will be discussed and compared to the previous experiment in the following section.

### 3.3 Assessment of the direct impact of IASI $\delta$D

To assess the direct impact of the assimilation of IASI $\delta$D on the meteorological analysis we performed an ensemble simulation with OSSE where only IASI $\delta$D is assimilated (called DA_IASI) and compare this simulation then to an IsoGSM ensemble simulation without any data assimilation (called noDA). This experiment is denoted in the following as "noDAvsDA" while the ensemble simulations where only conventional observations are assimilated and the one where IASI $\delta$D is assimilated additionally to the conventional observations (the two ensemble simulations discussed in the previous sections) are denoted as "PREPBUFR" (see Tab. 1)

Figure 7 shows bar charts of the skill for the noDAvsDA and the PREPBUFR experiment for the tropics and separated by regions in the troposphere (up to the 100 hPa level) derived from the vertical profiles for the month August 2016. As discussed in the previous section, for the PREPBUFR experiment an improvement in skill is found in the troposphere for all parameters for the entire tropics and for each sub-region when IASI $\delta$D is assimilated. For the noDAvsDA experiment we derive, irrespective if the entire tropics or specific regions in the tropics are considered, in the troposphere an improvement for all parameters except $\omega$, $Q_1$ and $Q_2$. For these three parameters a slight degradation ($-2$ to $-7\,\%$ for $\omega$, $-3$ to $-8\,\%$ for $Q_1$ and $-2$ to $-4\,\%$ for $Q_2$) is found (Fig. 7 and Tab. S1). However, this degradation is mainly restricted to the lowest or highest pressure levels (above 150-200 hPa and for $Q_1$ below 700 hPa, $Q_2$ below 500 hPa and $\omega$ below 800 hPa). Inbetween, thus in the free troposphere, the assimilation of $\delta$D has either no impact (skill around $0\,\%$) or causes a slight improvement (see Fig. S10). The improvement for the other parameters is in the range of 4-9 $\%$ for Asia, 14-22 $\%$ for America, 12-20 $\%$ for Africa and 11-17 $\%$ for the tropics).

Additionally to the improvement throughout the troposphere derived from the vertical profile we also consider time series. The time series have the advantage that it can be assessed how the assimilation experiment for a certain meteorological parameter at a certain altitude performs with respect to temporal variability of this parameter. Deriving the bar charts for the improvement in skill from the time series at the 500 hPa level (Fig. 8), the level where approximately the IASI data has been assimilated, we derive for the tropics an improvement for all parameters. Here, although very small, an improvement is also found for $\omega$, $Q_1$ and $Q_2$. Separated by regions, at this level, a slight degradation of temperature ($-3.14\,\%$) is solely found for Asia and in $Q_1$ and $Q_2$ for Africa ($-0.38$ and $-0.35\,\%$, so small that is hardly visible in Fig. 8, therefore see Tab. S4 instead). The improvement for the other parameters is in the range of 2-18 $\%$ for Asia, 3-27 $\%$ for America, 1-25 $\%$ for Africa and 2-21 $\%$ for the tropics (see Tab. S4). As from the profiles, also from the time series the lowest improvement is found for Asia for all parameters except the isotopologues. For these here the highest improvement is found (27.44 $\%$ for $\delta$D and 25.28 $\%$ for $\delta^{18}$O). The improvement in the precipitation rates (Fig. 8 and Tab. S4) is 3.97 $\%$ for Asia, 13.45 $\%$ for America and 7.79 $\%$ for Africa.

We generally derive from this experiment (noDAvsDA) similar results by region as for the PREPBUFR experiment, namely that in the troposphere the highest improvement is found for America. In contrast to the PREPBUFR experiment the lowest improvement is here found for Asia (Figure 7 and Tab. S2). However, for the degradation of $\omega$, $Q_1$ and $Q_2$ it is the opposite



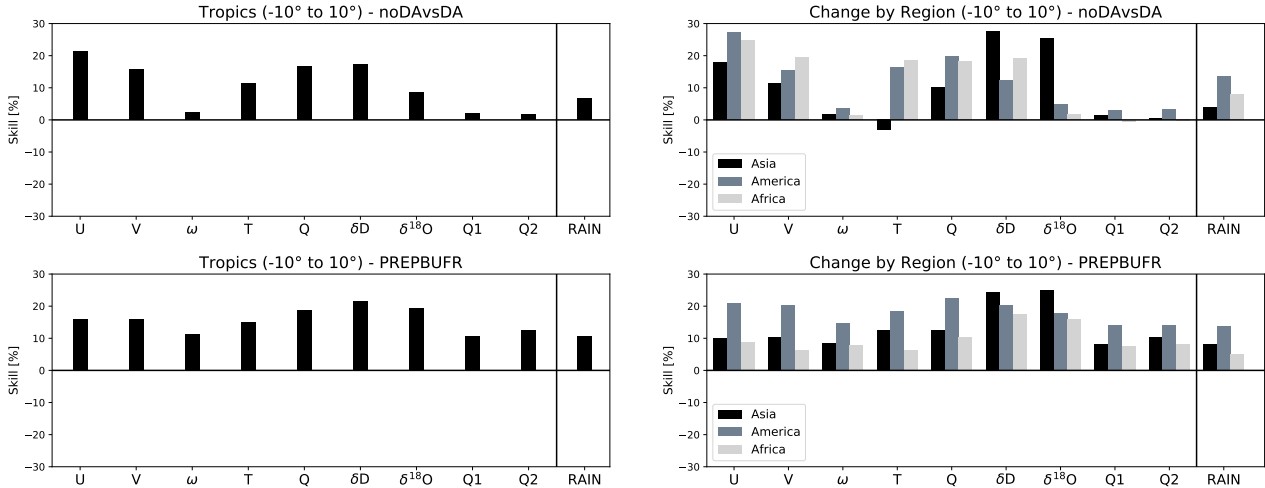

**Figure 8.** Same as 7, but derived from the time series at 500 hPa, except RAIN which is the precipitation accumulated at the surface level.

than for the improvement: The degradation is lowest for Asia. The highest degradation in $\omega$, $Q_1$ and $Q_2$ is found for Africa, although only slightly higher than for America (the average performance is about the same for these two regions). For the noDAvsDA experiment, we derive similar results from the time series at the 500 hPa level than from the vertical profiles,

namely that the lowest improvement is found for all parameters for Asia except for the isotopes. Here, as for the PREBUFR experiment, the highest improvement is found for America.

Figures 9 and 10 show a comparison of the time series of the mean difference between the ensemble mean of the respective assimilation run and the Nature run for the noDAvsDA and PREPBUFR experiment (6-hourly data for August 2016 at 500 hPa) for $Q_1$, $Q_2$ and $\omega$ and for the three regions considered here. Here, especially in the noDAvsDA experiment (Fig. 9) the positive

impact of assimilating IASI $\delta$D is quite obvious. While the noDA simulation lacks the synoptic-scale temporal variations that are found in the Nature run, the assimilation with IASI $\delta$D (DA_IASI) directly introduces these (see time series of the ensemble mean shown in Fig. S11). However, although these are not exactly the same as the one in the Nature run, especially for Asia and Africa, the agreement is quite reasonable.Therefore, the mean differences to the Nature run are for Asia and Africa larger for the assimilation run with IASI $\delta$D (DA_IASI). In contrast to these two regions, the DA_IASI assimilation is quite successful

for America. The agreement for $Q_1$, $Q_2$ and $\omega$ to the Nature run with just assimilating $\delta$D is impressive. This is reflected in the mean differences by lower differences for DA_IASI than for noDA which also alternate much closer around zero. The good agreement found here in the time series for America is in agreement with the results concerning skill discussed above, where we also found the highest improvement in skill for America.

In the case of the PREPBUFR experiment (Fig. 10 and Fig. S12), the assimilation of the conventional observations already

brings the analysis close to the Nature run and is further improved when IASI $\delta$D is additionally assimilated as can be seen from the RMSD and skill discussed a few paragraphs earlier (and shown in Fig. 8 and Tab. S3). In the mean differences this

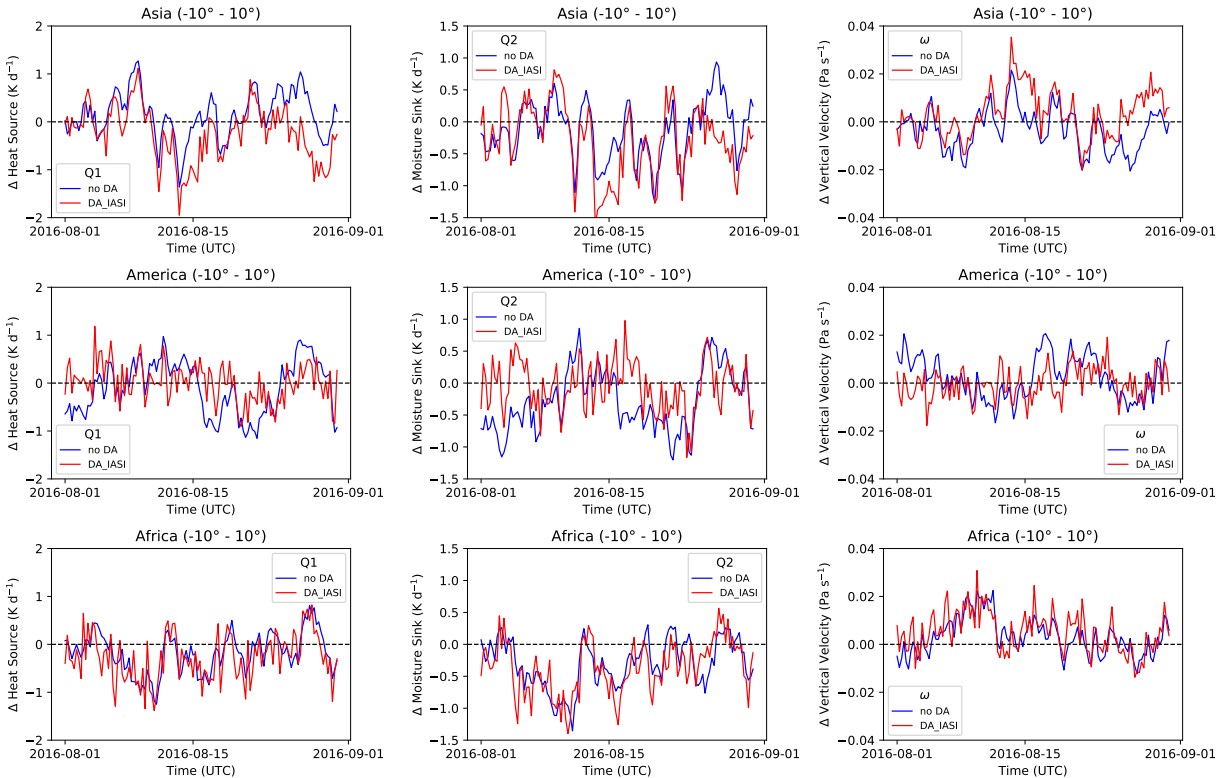

**Figure 9.** Time Series of the difference between assimilation run and Nature run for the heating source ($Q_1$), moisture sink ($Q_2$) and vertical velocity ($\omega$), from left to right) for Asia, America and Africa (from top to bottom) for the assimilation runs without any assimilation and with the assimilation of the mocked IASI data (noDAvsDA experiment) for the tropics ($10°$S to $10°$N) at $500\,\mathrm{hPa}$.

is reflected by lower differences from the Nature run for DA_IASI that alternate closer to zero than the assimilation run with conventional observations only.

## 3.4 Assessment by the $\delta$D-$\delta^{18}$O relationship and d-excess

Another example to demonstrate the benefit of the assimilation of IASI $\delta$D is when the relationship between $\delta$D and $\delta^{18}$O is considered. Figure 11 shows the correlation of $\delta$D and $\delta^{18}$O of the 6-hourly data for August 2016 at $500\,\mathrm{hPa}$ for the noDAvsDA (Fig. 11 top) and the PREPBUFR (Fig. 11 bottom) experiment at the three selected regions in the tropics. Additionally, the global and local meteoric water line (GMWL and LMWL, respectively) are shown. The GMWL is defined by:

$$\delta^{18}O = 8 \cdot \delta D + 10 \tag{7}$$



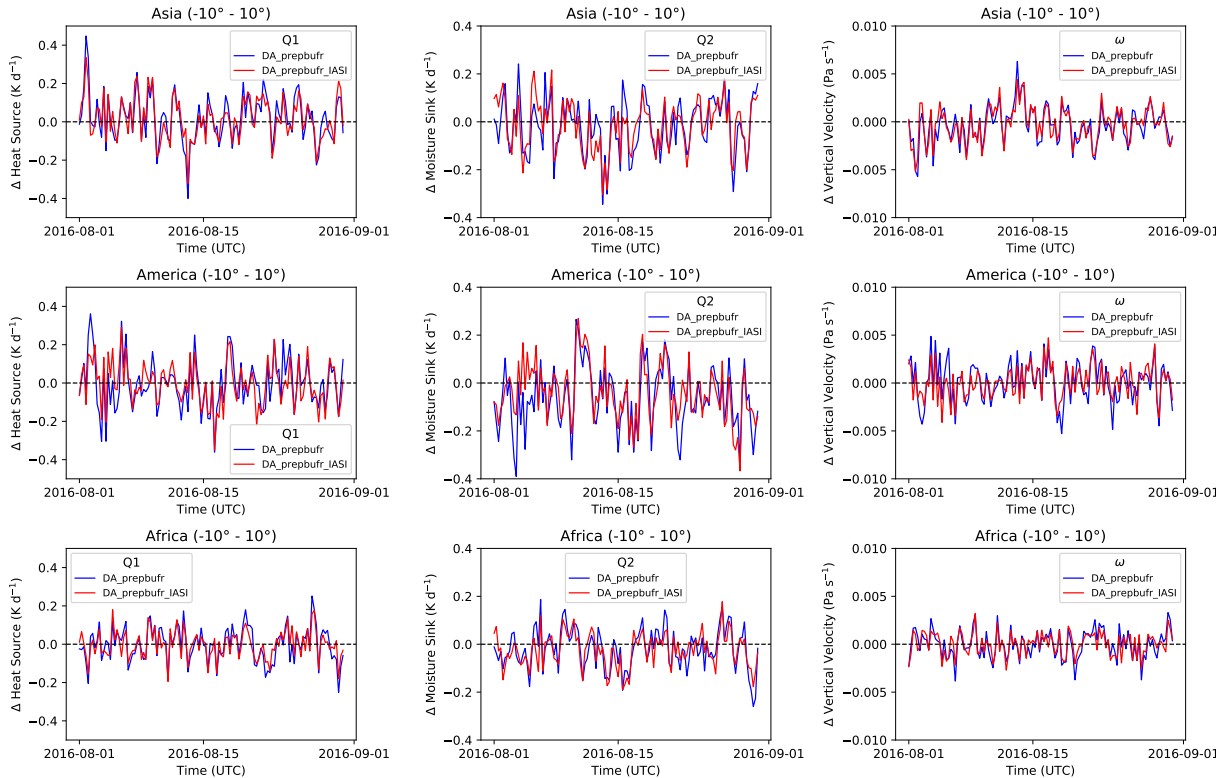

**Figure 10.** Same as Fig. 9, but here the time series are shown for the assimilation experiment with the conventional observations included (PREPBFUR experiment)

To derive the local meteoric water line (LMWL) for the here chosen areas a linear fit is applied to the correlation of $\delta D$ and $\delta^{18}O$ of the Nature run. We derive the following relationship for August 2016 at 500 hPa:

$$\delta^{18}O = 8 \cdot \delta D + 9 \quad \text{(Asia)}$$
$$\delta^{18}O = 8 \cdot \delta D + 20 \quad \text{(America)}$$
$$\delta^{18}O = 7 \cdot \delta D - 16 \quad \text{(Africa)} \tag{8}$$

Then $\delta^{18}O$ is calculated based on the $\delta D$ from the Nature and the assimilation runs, respectively. Figure 11 shows that the relationship between $\delta D$ and $\delta^{18}O$ is generally correct in IsoGSM, however, $\delta D$ and $\delta^{18}O$ show less variability and generally

a relationship with more depleted $\delta D$ and enriched $\delta^{18}O$. This deviation that is introduced by IsoGSM is most pronounced in Africa and America. The assimilation of IASI $\delta D$ alone helps to reduce this deviation and moves the correlation closer to the Nature. If the conventional observations are assimilated this deviation is also significantly reduced but an offset with respect to $\delta D$ and $\delta^{18}O$ still remains which can be corrected when additionally to the conventional observations IASI $\delta D$ is assimilated. For both, the PREPBUFR and noDAvsDA experiment, the best agreement in the $\delta D$-$\delta^{18}O$ relation is found for Asia which



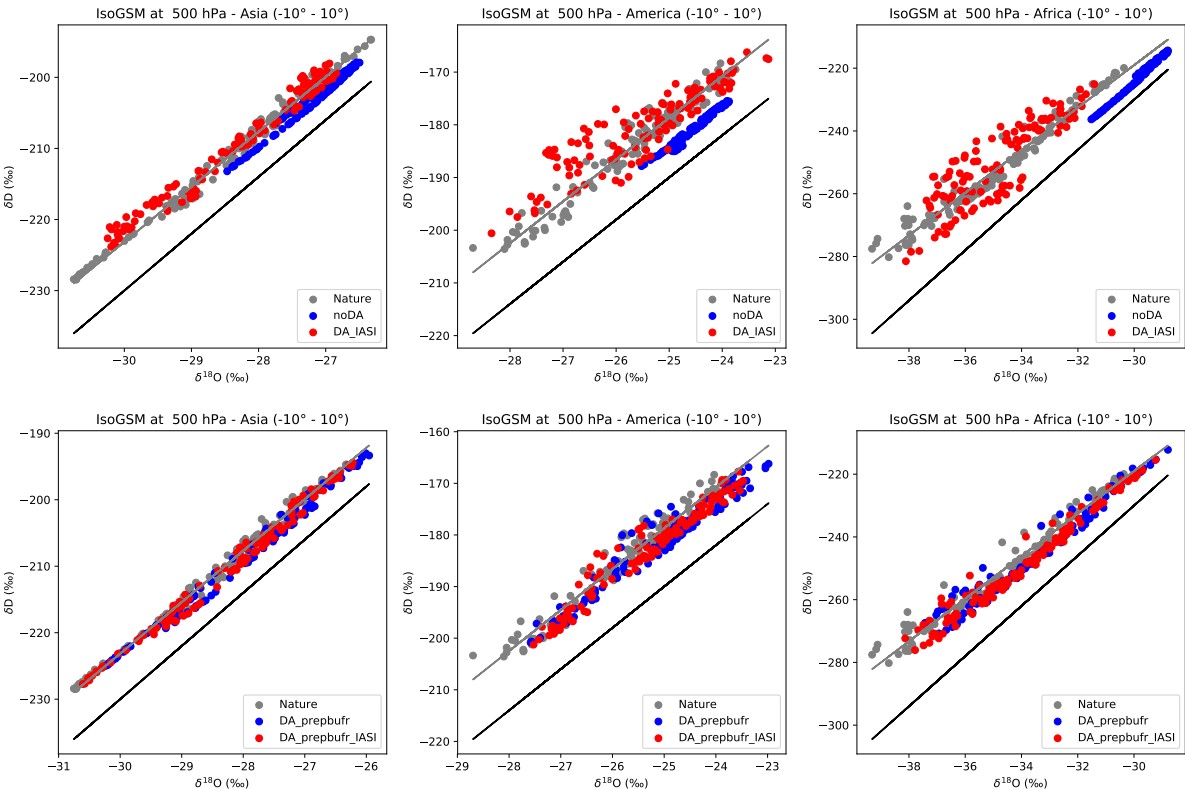

**Figure 11.** Correlation of $\delta^{18}$O vs $\delta$D for the noDAvsDA (top) and PREPBFUR (bottom) experiments for Asia, America and Africa at 500 hPa. The black line shows the Global Meteoric water line (GWML) and the grey the local meteoric water line (LMWL). Note, different x and y-scale ranges are used for the panels.

may be due to the fact that here the deviation in $\delta$D and $\delta^{18}$O is also lowest. For Africa and America, although the deviation can be significantly reduced, still a lot of scatter (noDAvsDA) and a slight offset (PREPBUFR) remains.

     Comparing the three regions with each other we find that Africa is the region which is most depleted (lowest $\delta$D and $\delta^{18}$O) and America is the region that is most enriched (highest $\delta$D and $\delta^{18}$O). The $\delta$D-$\delta^{18}$O correlation for Asia is quite similar to the one for America, but is by $\sim$40‰ more depleted than America. Further, the correlation for Africa spans over a much larger

value range than the correlation for America and Asia which is also reflected in a larger range of deuterium excess (d-excess) values (see Fig. 12). The second-order isotope variable d-excess (Daansgard, 1964) is a tracer for moisture source conditions and is also a measurable constraint for processes involved in precipitation formation (Aemisegger et al., 2015; Aemisgger and Sjolte, 2018). It is defined as follows: $d = \delta$D $- 8 \times \delta^{18}$O. While the d-excess for Asia agglomerates around 15‰, the d-excess for Africa spans over a much larger value range (17−40‰) with much more depleted (thus drier) $\delta^{18}$O than in the other two

regions. As larger the value range of the d-excess spans as larger also the differences between the assimilation experiments and the Nature run get.

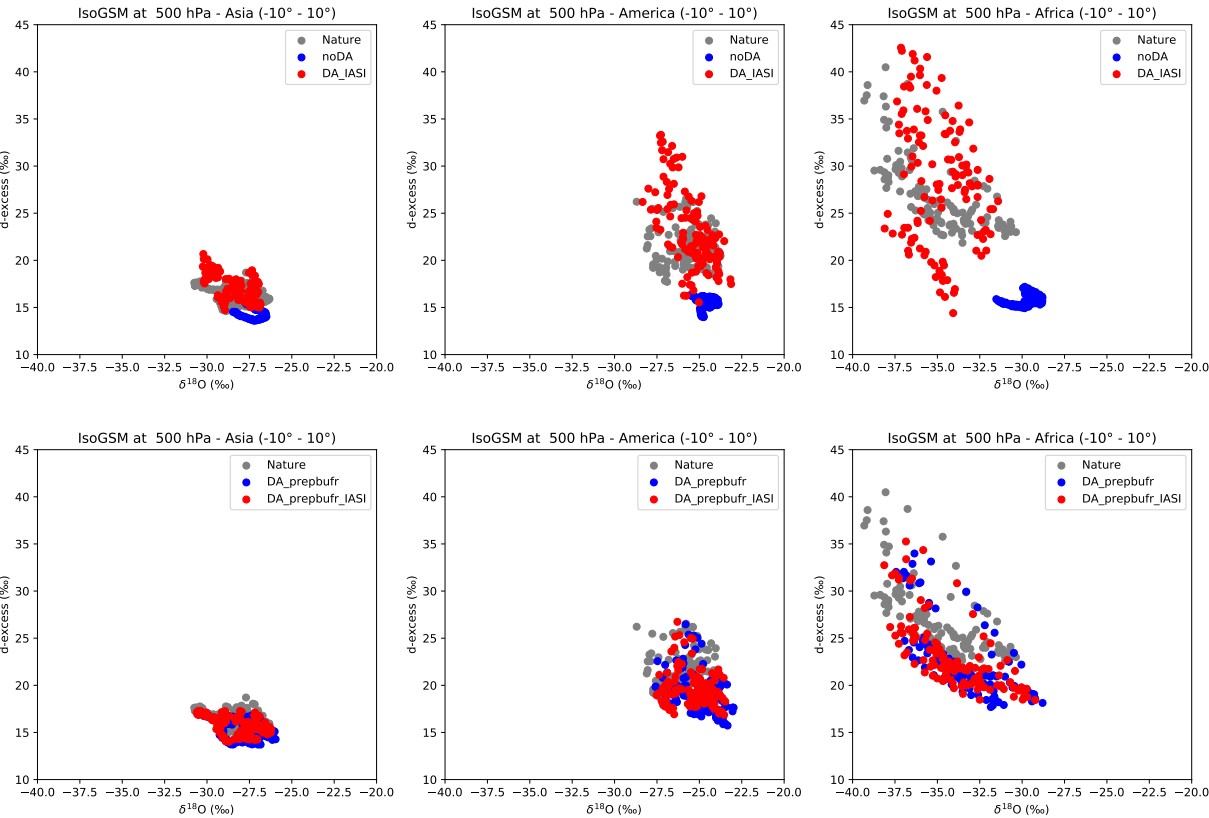

**Figure 12.** Correlation of $\delta^{18}$O vs d-excess for Asia, America and Africa at 500 hPa (top: noDAvsDA experiment, bottom: PREPBUFR experiment).

# 4  Discussion

The assimilation experiments described in the previous sections show that the assimilation of $\delta$D has the potential to improve the meteorological analysis in the tropics, both alone (noDAvsDA) and together with conventional observations (PREPBUFR).

However, heating rates and vertical motion can only be improved throughout the troposphere when additionally to IASI $\delta$D conventional observations are considered. When only IASI $\delta$D is assimilated the improvement in $\omega$, $Q_1$ and $Q_2$ is minor and restricted to the mid-troposphere. Further, there are differences in the performance of the assimilation experiments in the three regions in the tropics considered in this study (Asia, America and Africa). Thereby, we found the highest improvement for both experiments for America, while the lowest improvement was found for the PREPBUFR experiment for Africa and for the

noDAvsDA experiment for Asia.

High RMSDs of $\delta$D and $Q_2$ in the PREPBUFR experiment were found in the regions where the upward and downward branches of the atmospheric circulation are located (Fig. 3 and Fig. 4). Thereby, we found that these regions of high RMSD in $\delta$D coincide with regions where strong upward/downward motion and diabatic heating/cooling is dominant while for $Q_2$

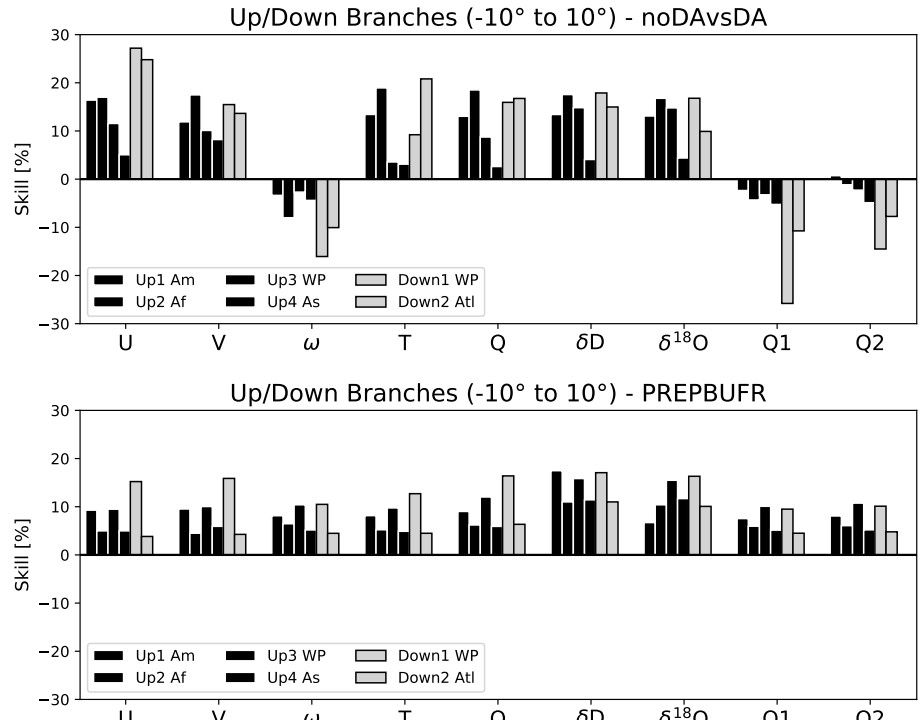

**Figure 13.** Improvement/degradation in skill in percent for each parameter in the troposphere (up to the 100 hPa level) derived from the vertical mean profiles and separated into upward and downward branches. Top: for the simulation runs with assimilation of the mocked IASI data compared to the simulation without any data assimilation (noDAvsDA experiment). Bottom: for the simulation with assimilation of the mocked IASI additional to the conventional observations compared to the simulation runs with only assimilating the conventional observations (PREPBUFR experiment).

these regions coincide with the regions where upward motion and heating is dominant. To investigate this relationship further,

additionally to the separation into the regions over land (Asia, America and Africa) analysed in the previous sections we made a separation into upward and downward branches and investigated the improvement in skill when IASI $\delta$D is assimilated in these regions (Fig. 13 and Fig. 14). To this end, we selected four upward branches (Up1 America (30°W–50°W), Up2 Africa (0°E–50°E, Up3 West Pacific (150°E–150°W and Up4 Asia 70°E–120°E)) and two downward branches (Down1 West Pacific (100°E–170°E) and Down2 Atlantic (50°W–10°E)).

We generally find, in agreement with Toride et al. (2021), an improvement in the atmospheric circulation when IASI $\delta$D is assimilated additionally to the conventional observations. However, for both experiments we do not see from the analyses performed here a better performance on either upward or downward branches. We rather find an improvement in the circulation cells dependent on region. For the PREPBUFR experiment, the highest improvement is found in the main Walker cell over the Pacific Ocean (130°E–150°W, Up3 WP and Down1 WP), which is only partly covered by the here defined Asian region.

Generally, the results derived from the analyses of the performance of the assimilation experiments in the upward/downward

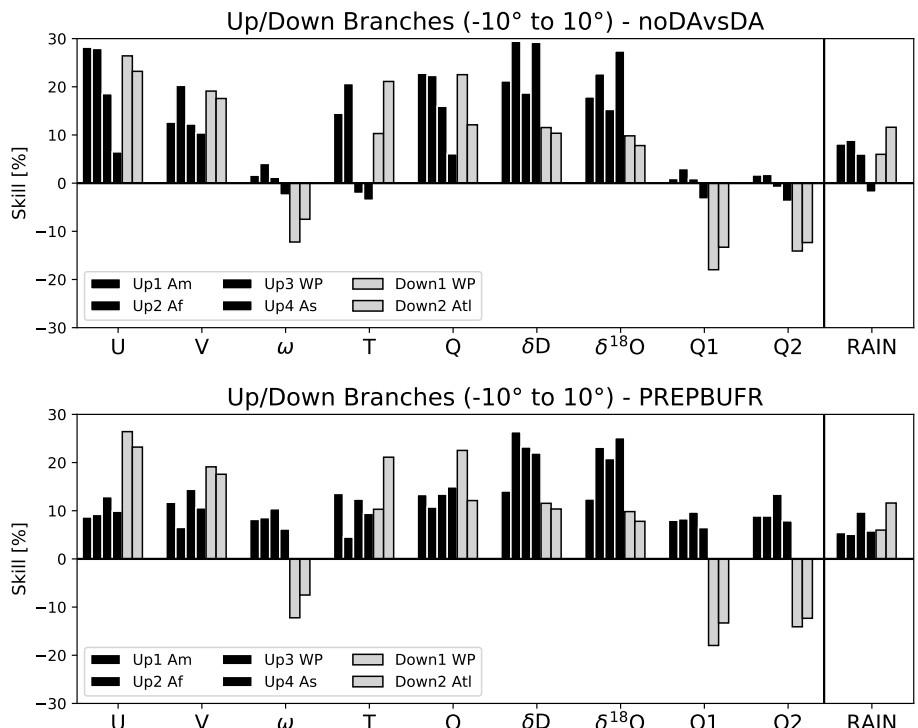

**Figure 14.** Same as 13, but derived from the time series at 500 hPa.

branches just confirms what we found before, namely that for both experiments the highest improvement is found for the respective upward/downward branches covered by the here defined American region and the lowest for the branches over the here defined African region (PREPBUFR) and Asian region (noDAvsDA). An exception are here $\omega$, $Q_1$ and $Q_2$ for which at the 500 hPa level better results are derived for the upward than for the downward branches.

We cannot entirely rule out why we see differences in the assimilation experiments by region, but a possible explanation may be the specific characteristics of these regions. The best agreement for both experiments was found for America, a region where moderate upward and downward motion prevails ($\omega$ alternating around zero), while the here defined Asian region is mostly dominated by strong upward motion and the African region by strong subsidence (Fig. 4 and Fig. S4). All three regions are affected by the respective monsoons, but in case of Asia and America the monsoon is located further north of our here 400 defined tropical region, while the monsoon over Africa is located at this time of the year directly over the equator and thus within the here defined region (Geen et al., 2020).

The above described intermediate behaviour of America is also reflected in the correlation of d-excess and $\delta^{18}$O (Fig. 12). While the d-excess for Asia agglomerates around 15‰ and the d-excess for Africa spans from 15-40‰, the d-excess for Africa lies in between, spanning the range from 15-25‰. In terms of the corresponding $\delta^{18}$O America is most enriched ($-27.5$ to 405 $-22.5$‰), while Asia and Africa are more depleted ($-27$ to $-30$‰ and $-27$ to $-40$‰, respectively). Further, considering precipitation rates, the lowest rates are found for Africa, while the highest are found for Asia (Fig. 15).



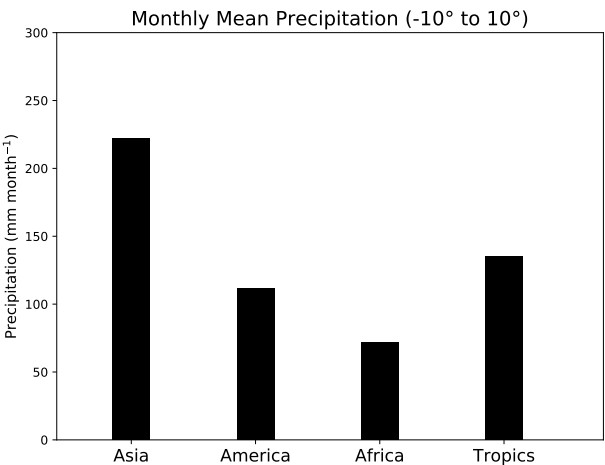

**Figure 15.** Monthly mean precipitation for the tropics and separated by region.

D-excess can act as fingerprints of earlier processes and thus high d-excess values can be associated with air that has been dried while low d-excess values can be associated with air that has been moistened (Salmon et al., 2019). Thus, the Asian regions which is mostly covered by oceanic areas is probably mostly affected by ocean evaporation (Risi et al., 2013).

Additionally with the above high precipitation rates the isotope ratios are in equilibrium resulting in the agglomeration of the d-excess at 15‰. America and Africa are mostly covered by land and are thus rather affected by continental recycling (Risi et al., 2013) and have intermediate and lower precipitation rates, respectively. This results in a deviation from the equilibrium and thus in a larger range of d-excess values showing that these regions are affected by both, moist and dry air.

Clear differences between the regions are also visible in the correlation of $\delta$D with $\delta^{18}$O (Fig. 11 and Eq. 8). While the slope

and intercept for Asia and America are indicating similar conditions (slope of 8 and a positive intercept), the correlation for Africa shows a different characteristic with the lower slope of 7 and an intercept that is negative (Putman et al., 2019). Further, comparing our mean profiles of $Q_2$ for these regions (Fig. 6) with the ones shown in Yokoyama et al. (2014) we find that during this time of the year (August 2016) Asia seems to be mostly affected by deep convection while America and Africa seem rather to be affected by shallow convection.

Considering all these differences we can conclude that we derive the best results for America because this region is affected by moderate dynamics (moderate up and downward motion with $\omega$ alternating around zero) than it is the case for Asia and Africa. In these two regions, dynamical processes are much stronger than in America. Especially, Asia which is quite humid and wet (high precipitation rates) is also a region where rather deep convection is prevailing (see also $\omega$ profile Fig. 6). This could explain, why the performance for the noDAvsDA experiment was lowest for Asia. Due to the underlying dynamics the

additional information from PREPBUFR is here more important than for the other two regions. Africa, on the other hand, with its complex dynamics and interaction between moist tropical and dry subtropical air masses and thus according effects on the isotopic composition seems to be generally an area that is difficult to simulate, both in terms of dynamics and isotopic





processes (e.g. Diekmann et al., 2021b). Thus, this may explain why we find the lowest improvement in this region for the
PREPBUFR experiment.

Nevertheless, besides the above discussed regional differences also the amount of data availability of the conventional observations and IASI $\delta$D in the respective areas for the assimilation experiments and the underlying model physics may definitely also play a role in the performance of the assimilation experiments in the three here considered tropical regions.

## 5   Conclusions

We performed idealized assimilation experiments where IASI $\delta$D data was mocked into an OSSE additional to conventional
observations (PREPBUFR). The assessment of the impact of this assimilation experiment on the meteorological analysis was performed for the tropics (10°S to 10°N). Thereby, additionally to the entire tropics also specific longitude regions in the tropics were considered, namely Asia (60°E to 180°E), America (120°W to 30°W) and Africa (30°W to 60°E).

    The assimilation experiment with IASI $\delta$D shows that for all parameters the RMSD can be decreased and the skill improved when the IASI $\delta$D data is assimilated additionally to the conventional observations (PREPBUFR experiment). The highest
improvement in skill and decrease in RMSD was found at ~500-600hPa, the approximate altitude where IASI $\delta$D has the highest sensitivity and was assimilated into the IsoGSM model. The improvement in skill for the PREPBUFR experiment is about 8-13 % for the tropical troposphere (up to the 100 hPa level). Separated by regions the improvement in the troposphere is about 7-13 % (Asia), 9-13 % for (America) and 4-10 % (Africa), respectively. Thus, the highest improvement is found for America and the lowest for Africa. Concerning the RMSD we found high RMSDs in $\delta$D, $Q_1$, $Q_2$ in certain regions. We found
that these regions of high RMSD in $\delta$D coincides with regions of upward/downward motion and heating/cooling while the high RMSD in $Q_2$ coincides with regions of upward motion and heating.

    In Addition to the PREPBUFR experiment, we performed another experiment consisting of an assimilation run where only IASI $\delta$D is assimilated and compared this to an IsoGSM ensemble simulation where no observations were assimilated to obtain the direct impact of the assimilation of IASI $\delta$D on the meteorological analysis fields (noDAvsDA experiment). In this
experiment we find an improvement in skill in the tropical troposphere for all parameters except $\omega$, $Q_1$ and $Q_2$ (from vertical profiles up to the 100 hPa level). However, the degradation for these parameters is restricted to the lowest atmospheric layers (below ~800 hPa). Above 800 hPa the improvement in skill is either around zero or slightly positive. The lowest improvement/highest degradation, thus lowest impact is found for Asia. From the time series at the 500 hPa level an improvement for all parameters is found except $T$ (Asia) and $Q_1$ and $Q_2$ (Africa) where a minimal degradation is found. Although also here the
lowest improvement is derived for Asia, for the isotopes it is here the opposite. Here, we find the highest improvement. From the vertical profiles, the highest improvement is found for all parameters for America and from the time series at the 500 hPa level for America when all parameters are considered and for Africa when only the parameters are considered where an improvement is found. The noDAvsDA experiment shows that the assimilation of IASI $\delta$D alone cannot significantly improve the heating rates. However, the assimilation of $\delta$D has a positive effect on all other parameters. Furthermore, together with the





conventional observations from PREPBUFR an additional improvement for all parameters, including the heating rates, can be achieved and shows the benefit of the IASI $\delta$D data.

Our study shows that the assimilation of IASI data has the potential (especially in combination with the conventional observations) to improve meteorological analysis and thus also weather forecasts and climate predictions. More promising results with OSSE can be derived if additionally to IASI $\delta$D also IASI $H_2O$ is assimilated (Toride et al., 2021). So far only idealized
experiments were performed, but experiments with assimilating real IASI $\delta$D data are in progress. However, a lot of uncertainties concerning water isotope modelling and observations remain (as discussed in Toride et al. (2021)) that could hinder the realisation of the assimilation of real IASI data and/or compared to the idealised experiments lessen the impact of $\delta$D on the analysis fields. Nevertheless, in the future, when vapor isotopic fields will be measured more frequently and the modelling of isotopic processes will be more accurate, the assimilation of isotopic observations may play an important role in improving the
analysis and forecast skill because isotopes provide unique information relevant to atmospheric circulation.

*Data availability.* The data of the assimilation experiments can be obtained from Zenodo (https://doi.org/10.5281/zenodo.4420315, Toride et al. (2021). The MUSICA IASI data are publicly available from the RADAR4KIT repository (https://doi.org/10.35097/408, Schneider et al., 2021 and https://dx.doi.org/10.35097/415, Diekmann et al. (2021a))

*Author contributions.* This study was designed by FK, MS, KY and KT. KY and KT performed the assimilation experiments. FK analysed
the assimilation experiments and wrote the manuscript with input from all co-authors. MS, CD, BE retrieved and provided the IASI data.

*Competing interests.* The authors declare that they have no conflict of interest.

*Acknowledgements.* This work was funded by the DFG project TEDDY (416767181, Geschäftszeichen SCHN 1126/5-1). We also acknowledge funding from the DFG project MOTIV (290612604, Geschäftszeichen SCHN 1126/2-1). The MUSICA IASI processing is performed on the supercomputer ForHLR funded by the Ministry of Science, Research and the Arts Baden-Württemberg and by the German Federal
Ministry of Education and Research.

The article processing charges for this open-access publication were covered
by a Research Centre of the Helmholtz Association.



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
