# Peer review of "Can the assimilation of water isotopologue observation improve the quality of tropical diabatic heating and precipitation?"

_Weather and Climate Dynamics, 2021_

## Referee Comment (RC1)

**Review of Khosrawi et al**

September 9, 2021

This paper documents the improvements in the simulation of meteorological fields in the tropics through the assimilation of synthetic isotopic observations. The added value compared to Toride et al 2020, a previous paper by the same group, is not clear (major comment 1.1). The paper is long, giving the impression of new information compared to Toride et al, but this is because the paper includes many paragraphs that are not clearly connected with the subject (major comment 1.2).

Another major problem is that the added value of the assimilation of isotopic observations compared to simpler humidity observations is not clear, and not even discussed. I fear that readers of this paper could be mislead about the advantages of assimilating water isotopic observations (major comments 1.3).

There is also a problem with the definition of the regions of interest, which seems to stem from misunderstanding of the global precipitation distribution and seasonal migration of the ITCZ (major comment 1.4).

**1 Major comments**

**1.1 The added value of this paper compared to Toride et al 2020 is not clear**

Toride et al already documents the added value of assimilating water isotopic observations to improve the simulation of atmospheric variables. l 59-63, the authors justify the added value of this study compared to Toride using two arguments:

- Toride et al was global whereas this paper focuses on the tropics. Yet, in Toride et al, there were already maps and cross sections allowing us to see what happens in the Tropics.

- This paper focuses on latent heating profiles. However, Toride et al already included discussion of the effects on latent heating profiles. The vertical circulation and the precipitation , which are documented in this study, were already documented in Toride et al as well.

In addition, the discussion of latent heating and vertical circulation in this study is mainly restricted to a description of the spatial fields, which have already been known for a long time and do not tell anything about the added value of isotopic assimilation (see major comment 1.3).

Therefore, the added value of this paper compared to Toride et al is not clear.

A possible added value, which could be interesting, is to try and understand the mechanisms by which the assimilation of $\delta D$ improves the latent heating profiles. The improvements described in the paper could be physically explained.

**1.2 The paper includes many paragraphs that are not clearly connected with the subject**

- p 3-4: 18 lines are devoted to the description of the IASI observations, yet, these are not used in this study. Rather, synthetic data mimicking IASI observations are used. This sub-section should be reoriented to describe the generation of the synthetic IASI data. A brief description of the IASI observations can be useful with this aim, but not per se, and only the information useful to generate the synthetic data is necessary.

- l 235-265: this describes the vertical circulation and diabatic profiles in 3 regions. These are well-known features of the large-scale atmospheric circulation. This could be recalled in just a few lines, with adequate citations. In addition, the differences between the regions simply reflect the summer location of the ITCZ,

not intrinsic properties of convection over different continents as suggested by the paper (see comment 1.4).

- l 357-366: the connection between this discussion and the effect of assimilation is not clear.

- Fig 15: this is very basic climatology and could go to SI. Again, this simply reflects the ITCZ location relative to the defined boxes (see comment 1.4).

- l 407-419: the connection with the subject of the paper is not clear.

The paper could be much shorter if it was more focused on its initial science question.

**1.3 The added value of the assimilation of isotopic observations compared to simpler humidity observations is not clear**

The water isotopic composition is often strongly correlated with the specific humidity in both observations and models, e.g. [Noone, 2012, Galewsky et al., 2016]. Measuring specific humidity is much easier, cheaper and widespread than measuring the isotopic composition. Therefore, the information gained from isotopic observations is always assessed relative to the information gained from specific humidity.

Here, the study quantified the improvement associated with the assimilation of isotopic observations, but what is the part of this improvement that we could already have just by assimilating humidity observations? Toride et al showed that actually, the improvement would be even better if assimilating humidity observations than if assimilating isotopic observations, and that the improvement is tiny if assimilating both $\delta D$ and humidity compared to assimilating only humidity.

Therefore, I fear that readers of this paper could be mislead about the advantages of assimilating water isotopic observations. I think that the added value of assimilating isotopic observations should be quantified relative to assimilating both conventional variables and concomitant humidity observations, and not just relative to conventional measurements.

**1.4 Problem with the definition of regions of interest**

Why selecting the 10S-10N region, for a month of August? It is well-known that the ITCZ during this month is located further North. Fig 5 illustrates that the defined regions are on the edge of the ITCZ. These boxes are thus heterogeneous, with one part in the ITCZ and one outside. The analysis would be more meaningful if the boxes represented more homogeneous meteorological conditions.

What is the rationale for choosing these regions? If the goal is to look at the ITCZ, then boxes further North should be chosen. If the goal is to look at the descending branches of the Hadley cell, then boxes should be chosen further South.

I suspect that the definition of the regions actually stems from mis-understanding of the global precipitation distribution and of the seasonal migration of the ITCZ. This is reflected by the wrong statement in l 400: "the monsoon over Africa is located at this time of the year directly over the equator". It has long been observed that the ITCZ over Africa in summer is located around 10N, as shown by your Fig 5 and documented by many previous studies, e.g. [Thorncroft et al., 2011]. The authors cite Geen et al 2020 for this statement, yet that also show seasonal monsoons in Africa with the ITCZ located around 10N in summer (see their fig 1).

If boxes were adequately chosen, the additional separation into upward and downward branches (l 376-394) would become useless, and this would additionally simplify the paper (see comment 1.2).

**2 Minor comments**

- l 217: "the absence of convection ... leads to strong subsidence": or rather vis versa?

- l 325: "lacks the synoptic-scale temporal variations": these are not visible on Fig 9. Fig 9 should rather be plotted in absolute values, not differences. Otherwise, we cannot see what is the magnitude of the synoptic-scale temporal variations. l 305: "with respect to temporal variability of this parameter": this is an additional reason why Fig 9 should show the temporal variability, not just differences between simulations.

- l 349: "correct in isoGSM": correct to what observations? No observations are shown here.

- Fig 11: why is the noDA simulation so difference from the Nature run? What is the difference between the noDA and the Nature run, except for different initial conditions? I can understand different synoptic variations, but why such different mean $\delta D$-$\delta^{18}O$ relationships?

- l 365: this sentence is not grammatically correct.

- When $\delta D$ is assimilated, what happens with $\delta^{18}O$? Is it left free? Or is it assimilated assuming some $\delta D$-$\delta^{18}O$ relationship? What is the impact of the way $\delta^{18}O$ is assimilated (or not) on the results regarding $\delta D$-$\delta^{18}O$ relationships and d-excess?

- l 408: wrong, moistening by rain evaporation is known to increase d-excess in the vapor, e.g. [Tremoy et al., 2014]

- l 445: upward or downward? heating or cooling? If both, this sentence should just say "all regions"

**References**

[Galewsky et al., 2016] Galewsky, J., Steen-Larsen, H. C., Field, R. D., Worden, J., Risi, C., and Schneider, M. (2016). Stable isotopes in atmospheric water vapor and applications to the hydrologic cycle. *Reviews of Geophysics*, 54(4):809–865.

[Noone, 2012] Noone, D. (2012). Pairing measurements of the water vapor isotope ratio with humidity to deduce atmospheric moistening and dehydration in the tropical mid-troposphere. *Journal of Climate*, 25(13):4476–4494.

[Thorncroft et al., 2011] Thorncroft, C. D., Nguyen, H., Zhang, C., and Peyrillé, P. (2011). Annual cycle of the west african monsoon: regional circulations and associated water vapour transport. *Quarterly Journal of the Royal Meteorological Society*, 137(654):129–147.

[Tremoy et al., 2014] Tremoy, G., Vimeux, F., Soumana, S., Souley, I., Risi, C., Cattani, O., Favreau, G., and Oi, M. (2014). Clustering mesoscale convective systems with laser-based water vapor delta18O monitoring in Niamey (Niger). *J. Geophys. Res.*, 119(9):5079–5103, DOI: 10.1002/2013JD020968.

---

## Referee Comment (RC2)

The aim of this paper is to show the benefit of stable water isotope observation assimilation for improving the representation of diabatic heating and precipitation in the tropics. A theoretical approach is chosen based on Observation System Simulation Experiments (OSSEs). The OSSEs are nearly the same as the ones presented earlier this year in Toride et al. 2021. While I do think that water isotopes contain valuable additional information on atmospheric circulation characteristics and moist diabatic processes in the atmosphere, I am very skeptical about their direct usefulness in data assimilation. In my view, there is no evidence provided in this paper that would support such a conclusion. The major reasons, why I think that the paper is difficult to understand in the current form are:

1) **Contradiction in stated hypothesis of the physical reason for the added value of isotopes in data assimilation and the outcome of the second OSSE experiment**
As stated by the authors in the introduction, the rationale for the use of isotope observations to improve various meteorological fields such as $T,q,u,v$ is that they are tracers of moist diabatic processes in the atmosphere. Thus, via improvements in diabatic heating rates in models, isotope assimilation leads to improvements in other fields. However, that is not what the authors observe in their second OSSE, in which they only assimilate $\delta D$. In the noDavsDa experiment the authors find an improvement in all variables except those *($\omega$, Q1, Q2)*, for which we would expect a direct physical link with the mid tropospheric $\delta D$ distribution to exist. This contradiction is very disturbing for the readers and unfortunately not addressed at all by the authors. Based on this result, what do the authors think, is the reason for the improvements observed in the other meteorological fields?

2) **Observation density**
Since $\delta D$ assimilation can only lead to substantial improvements in diabatic heating when assimilated together with conventional observations, the question about the observation density arises. This should be discussed and an assessment of the observation density differences in the PREBUFR experiments should be provided. I know that this is done in the supplement of Toride et al. 2021, but I think this is so essential that it cannot just be left out of the discussion in this paper. Increasing the number of conventional observations at the locations of assimilated IASI $\delta D$ (e.g. *q* profiles from IASI) instead of $\delta D$ would maybe lead to even larger improvements.

3) **Motivation for chosen tropical region delimitation**
I missed a clear motivation for the chosen tropical regions, over which the $\delta D$ induced improvements in data assimilation are quantified. Why not focusing on known ascent dominated regions along the ITCZ vs. subsidence dominated regions further away from the equator? In the current form I did not gain any process-based insight from the regional categorization.

4) **Missing discussion on precipitation improvements**
Even though improvements in modelled precipitation seem to be expected through improvements in diabatic heating profiles, I find the discussion about precipitation too sparse to allow for such a prominent place in the title.

Minor comments:
- Many parts of the paper are a bit lengthy in writing and in the shown Figures. For example:
  o A lot of information is given about IASI, even though no real IASI data is used
  o I cannot see the differences in the profiles shown in Fig. 6.
  o What can I learn from Figures 9 and 10?
  o The role of Section 3.4 about the $\delta$D-$\delta^{18}$O relation and dexcess is not clear to me and does not fit well into the storyline.
- I did not understand the difference between the individual ensemble members. Were they just initialized at different times from the nature run? If yes, why are they different from the nature run, then? Or are the initial conditions perturbed with respect to the nature run?

---

## Author Comment (AC1)

**Reply to Referee 1 Comments**

**Manuscript-No: wcd-2019-49**

**Can the assimilation of water isotopologue observation improve the quality of tropical diabatic heating and precipitation?**

**We thank referee 1 for the constructive, helpful criticism and the suggestion for revision. We have thoroughly revised the manuscript based on the comments given by the referees. A detailed point-by-point response to the comments by referee 1 are given below.**

*This paper documents the improvements in the simulation of meteorological fields in the tropics through the assimilation of synthetic isotopic observations. The added value compared to Toride et al 2020, a previous paper by the same group, is not clear (major comment 1.1). The paper is long, giving the impression of new information compared to Toride et al, but this is because the paper includes many paragraphs that are not clearly connected with the subject (major comment 1.2). Another major problem is that the added value of the assimilation of isotopic observations compared to simpler humidity observations is not clear, and not even discussed. I fear that readers of this paper could be mislead about the advantages of assimilating water isotopic observations (major comments 1.3). There is also a problem with the definition of the regions of interest, which seems to stem from misunderstanding of the global precipitation distribution and seasonal migration of the ITCZ (major comment 1.4).*

**We apologize for not having been clear in our manuscript which seems to have caused some confusion. We are focusing in our study on the Walker circulation which spans in east-west direction over the entire tropics. This is why we chose the latitude band between 10°S to 10°N. Further, there are clear differences between the study by Toride et al. (2021) and our study as described below. This will be better pointed out in the revised manuscript. Furthermore, it was not our intention to mislead the readers on the advantages of assimilating isotopic observations. We will also make this point more clear in the revised manuscript and will discuss the possibilities and limitations.**

*1 Major comments*
*1.1 The added value of this paper compared to Toride et al 2020 is not clear Toride et al already documents the added value of assimilating water isotopic observations to improve the simulation of atmospheric variables. l 59-63, the authors justify the added value of this study compared to Toride using two arguments:*

**The main difference between our study and the study by Toride et al. (2021) is that Toride et al. (2021) have their main focus on the assessment of the assimilation experiments using IASI $\delta$D, IASI water vapour or or both on a global scale and on the separation into thermodynamic and dynamic contributions while we try to assess the direct impact that the assimilation of $\delta$D has on the meteorological analyses and especially on the diabatic heating rates in the tropics. We therefore compare the assimilation experiment with IASI $\delta$D additional to conventional observations (PREPBUFR experiment) from Toride et al. (2021) to one assimilation experiment that has been performed in the frame of this study where only IASI $\delta$D is assimilated without any other observations (noDAvsDA experiment). We select three specific regions in the tropics and additionally separate the tropics into the upward and downward branches of the Walker circulation to assess the performance of the assimilation experiments there. Additionally, we consider the $\delta$D-$\delta^{18}O$ relationship and the d-excess which serves on one hand as a further tool for**

assessing the performance of the assimilation experiments and on the other hand helps us to explain the differences in performance for three regions considered. Therefore, there are clear differences between our study and the study by Toride et al. (2021). We hope that the changes we made in the introduction (the 5th and 6th paragraph has been rewritten) make the differences between the two studies more clear.

- *Toride et al was global whereas this paper focuses on the tropics. Yet, in Toride et al, there were already maps and cross sections allowing us to see what happens in the Tropics.*
  **The study by Toride et al. (2021) is mainly showing global results. The global maps shown in Toride et al. are only for one altitude level, namely for 500 hPa. Thus, there one only can see what happens (compared to other regions on the globe) at 500 hPa. In our study we solely focus on the tropics (especially the inner tropics) and assess the performance in this region by using profiles (assessing the performance over the entire troposphere) and time series (assessing the performance over one month considering the temporal fluctuations). The longitude-pressure cross sections shown by Toride et al (2021) are for the circulation cells between 0-30°N and a specific longitude range in the tropics while we show and consider in our study the inner tropics and all circulation cells of the Walker circulation spanning over the entire tropics, separate the tropics in three regions and consider the performance there and also separate by upward and downward branches of the Walker circulation.**

- *This paper focuses on latent heating profiles. However, Toride et al already included discussion of the effects on latent heating profiles. The vertical circulation and the precipitation, which are documented in this study, were already documented in Toride et al as well.*
  **In Toride et al. (2021) only latent heating is shown for the longitude-pressure cross section and the improvement for the circulation is discussed in a specific longitude range for the latitude band 0 to 30°N while we discuss the latent heating for the inner tropics (10°S to 10°N) and for the three regions within the tropics based on vertical profiles (average improvement over the troposphere), cross sections (all longitudes, thus the entire Walker circulation) and time series (temporal fluctuations). Precipitation results in Toride et al. (2021) are only shown on a global map and in the results for the forecast while we here asses the improvement in precipitation based on the time series for the inner tropics and separated by regions.**

**To make the differences between our study and the study by Toride et al. (2021) more clear, the 5th and 6th paragraph of the introduction have been rewritten.**

*In addition, the discussion of latent heating and vertical circulation in this study is mainly restricted to a description of the spatial fields, which have already been known for a long time and do not tell anything about the added value of isotopic assimilation (see major comment 1.3).* **We apologize that we could not get the message through. Our intention with showing and discussing the vertical profiles and the longitude-pressure cross sections was to describe and explain the differences between the regions. These are then in the end (in the discussion) used to explain why we find differences in the performance and why we have problems with deriving an improvement in heating rates and vertical velocity when only isotopologues are assimilated without any other data (noDAvsDA experiment).**

*Therefore, the added value of this paper compared to Toride et al is not clear.*
*A possible added value, which could be interesting, is to try and understand the mechanisms by*

*which the assimilation of $\delta D$ improves the latent heating profiles. The improvements described in the paper could be physically explained.*

**This is actually what we tried. Our experiments show that diabatic heating and vertical motion can only be improved when the underlying physics (dynamics) are already given through the assimilation of conventional observations (the PREPBUFR data and our PREPBUFR experiment, respectively). If only isotopes are assimilated without any other data we get the atmospheric variability to some extent right (the ensemble mean has little variability and thus the mean reflects the climatological conditions), but not the exact fluctuations which results in small or no improvements compared to the Nature run when quantitatively assessed using the skill. This effect and the qualitative improvement is clearly visible in the time series (Fig. S11 and S12 in the supplement).**

*The paper includes many paragraphs that are not clearly connected with the subject*

- *p 3-4: 18 lines are devoted to the description of the IASI observations, yet, these are not used in this study. Rather, synthetic data mimicking IASI observations are used. This sub-section should be reoriented to describe the generation of the synthetic IASI data. A brief description of the IASI observations can be useful with this aim, but not per se, and only the information useful to generate the synthetic data is necessary.*
  **This is definitely true. We followed the suggestion by the referee and removed the IASI subsection and moved the parts that are needed for the description of the generation of the synthetic data to the section where we describe the OSSE (former section 2.3, now section 2.2).**

- *l 235-265: this describes the vertical circulation and diabatic profiles in 3 regions. These are well-known features of the large-scale atmospheric circulation. This could be recalled in just a few lines, with adequate citations. In addition, the differences between the regions simply reflect the summer location of the ITCZ, not intrinsic properties of convection over different continents as suggested by the paper (see comment 1.4).*
  **We have shortened this paragraphs as suggested. However, please note, that we here focus on the Walker circulation and not on the ITCZ. The cross sections we show are longitudinal cross sections, not latitudinal cross sections. To make this more clear throughout the manuscript we have added the suffix "longitude-pressure" before each mentioning of the "cross sections".**

- *l 357-366: the connection between this discussion and the effect of assimilation is not clear.*
  **We use the $\delta D$-$\delta^{18}O$ relationship and d-excess to understand the differences in isotopic processes between the three regions considered in the assessment of the assimilation experiments. We try to relate that back to why in one region the assimilation of $\delta D$ is more successful than in an other region. We have improved this section and hope that this becomes now more clear. Additionally, we added a sentence in the introduction (where we describe the structure of the paper) to motivate the application of the $\delta D$-$\delta^{18}O$ relationship and d-excess.**

- *Fig 15: this is very basic climatology and could go to SI. Again, this simply reflects the ITCZ location relative to the defined boxes (see comment 1.4).*
  **We agree that this figure could be moved to the supplement and thus followed this suggestion.**

- *l 407-419: the connection with the subject of the paper is not clear. The paper could be much shorter if it was more focused on its initial science question.*
  **As mentioned above the $\delta$D-$\delta^{18}$O relationship and d-excess are used to understand the differences in isotope processes between the three regions considered in the assessment of the assimilation experiments. We have revised this section to make our intention clearer.**

*1.3 The added value of the assimilation of isotopic observations compared to simpler humidity observations is not clear*

**Toride et al. (2021) and also our study shows that the assimilation of IASI isotopes alone leads to an improvement although the assimilation of IASI water vapour or both, water vapour and $\delta$D, is more efficient. Further, our main intention was to investigate which information is stored in the isotope data and how the isotopes alone can improve diabatic heating, which is one of the main differences between our study and the study by Toride et al. (2021). The aim of our study was not to present the assimilation experiment that provides the highest improvement. We however agree that we need to make throughout the manuscript more clear that better results in terms of improvement can be derived when IASI water vapour or even both are assimilated.**

*The water isotopic composition is often strongly correlated with the specific humidity in both observations and models, e.g. [Noone, 2012, Galewsky et al., 2016]. Measuring specific humidity is much easier, cheaper and widespread than measuring the isotopic composition. Therefore, the information gained from isotopic observations is always assessed relative to the information gained from specific humidity.*

**Isotopic observation are becoming more and more available in the future. Especially with the IASI instruments on the coming generations of Metop satellites will make $\delta$D observations with a very high resolution available for the next decades. Further, our intention is to investigate which information the isotopes hold, thus we solely focus in our study on these. Although isotopes are correlated with specific humidity and this relationship is quite valuable for understanding isotopic processes, one does not necessarily needs to exploit this relationship. This definitely depends on the intention of the study. Further, Toride et al. (2021) confirmed with their assimilation experiments what is discussed in Galewsky et al. (2016) (and references therein), that isotopes hold different information than specific humidity (the IASI $\delta$D - q vs IASI-q experiment).**

*Here, the study quantified the improvement associated with the assimilation of isotopic observations, but what is the part of this improvement that we could already have just by assimilating humidity observations? Toride et al showed that actually, the improvement would be even better if assimilating humidity observations than if assimilating isotopic observations, and that the improvement is tiny if assimilating both $\delta D$ and humidity compared to assimilating only humidity.*
**We agree that this should be better pointed out throughout the manuscript and improved the text accordingly. Nevertheless, even though the assimilation of water vapour is more efficient this does not mean that the assimilation of IASI $\delta$D is worthless.**

*Therefore, I fear that readers of this paper could be mislead about the advantages of assimilating water isotopic observations. I think that the added value of assimilating isotopic observations should be quantified relative to assimilating both conventional variables and concomitant humidity observations, and not just relative to conventional measurements.*

**Since specific humidity is not assimilated in PREPBUFR to the extent (spatial and temporal resolution) as provided by the IASI instrument and due to the fact that PREPBUFR is the current state of the art assimilation data set, we think it is more correct to make the assessment relative to this data set to show what one would gain when additionally to PREPBUFR IASI data is assimilated, irrespective if water vapour or isotopes. Further, our intention is to investigate the direct impact of the isotopologue assimilation, thus which information the isotopes hold. Therefore, we solely focus in our study on these.**

*1.4 Problem with the definition of regions of interest*
*Why selecting the 10S-10N region, for a month of August? It is well-known that the ITCZ during this month is located further North. Fig 5 illustrates that the defined regions are on the edge of the ITCZ. These boxes are thus heterogeneous, with one part in the ITCZ and one outside. The analysis would be more meaningful if the boxes represented more homogeneous meteorological conditions.*
**Our major focus is on diabatic heating and the Walker circulation. Studies focusing on this also used the latitude band between 10°S-10°N or even a smaller latitude band of 5°S-5°N (e.g. Fueglistaler et al., 2009; Wright and Fueglistaler, 2013; Dee et al., 2018). Nevertheless, we repeated our assessment for the tropics and separated by regions (average improvement from the profiles) for different latitude bands (20°S to 20°N, 0° to 20°N and 20°S to 0°) to show that our results are robust and do not depend on the chosen tropical latitude band (Figure 1 and Figure 2 in this reply).**

*What is the rationale for choosing these regions? If the goal is to look at the ITCZ, then boxes further North should be chosen. If the goal is to look at the descending branches of the Hadley cell, then boxes should be chosen further South.*
*I suspect that the definition of the regions actually stems from mis-understanding of the global precipitation distribution and of the seasonal migration of the ITCZ. This is reflected by the wrong statement in l 400: "the monsoon over Africa is located at this time of the year directly over the equator". It has long been observed that the ITCZ over Africa in summer is located around 10N, as shown by your Fig 5 and documented by many previous studies, e.g. [Thorncroft et al., 2011]. The authors cite Geen et al 2020 for this statement, yet that also show seasonal monsoons in Africa with the ITCZ located around 10○N in summer (see their fig 1).*
**Our focus is not on the ITCZ and Figure 5 shows a longitudinal cross section. Our point here was that in contrast to the other two regions we get some influence from the monsoon in the here considered African region. According to the comment by the referee, we changed the sentence as follows to be more precise and also added the reference of Thorncroft et al. (2011): "All three regions are affected by the respective monsoons, but in case of Asia and America the monsoon is located further north of our here defined tropical region, while the monsoon over Africa is located at this time of the year at around 10°N and thus within the here defined region (e.g. Thorncroft et al.2011; Geen et al., 2020)."**

*If boxes were adequately chosen, the additional separation into upward and downward branches (l 376-394) would become useless, and this would additionally simplify the paper (see comment 1.2).*
**Even if we would move our boxes further north or south, thus using a different latitude band, we would still have different upward and downward branches and a separation into**

**these would be useful (see Fig. 3 in this reply).**

*2 Minor comments*

- *l 217: "the absence of convection ... leads to strong subsidence": or rather vis versa?*
  **You are right, thanks for pointing this out. We changed the sentence as folllows: "Conversely, the strong subsidence over the eastern Pacific leads to the absence of convection (Wright et al. 2013)".**

- *l 325: "lacks the synoptic-scale temporal variations": these are not visible on Fig 9. Fig 9 should rather be plotted in absolute values, not differences. Otherwise, we cannot see what is the magnitude of the synoptic-scale temporal variations. l 305: "with respect to temporal variability of this parameter": this is an additional reason why Fig 9 should show the temporal variability, not just differences between simulations.*
  **We had provided the figures showing the absolute values in the supplement, but based on your comment we now swapped the figures and put the one with the absolute values for the noDAvsDA experiment in the manuscript and the ones with the differences in the supplement.**

- *l 349: "correct in isoGSM": correct to what observations? No observations are shown here.*
  **With this statement we just meant, that we derive the expected linear $\delta$D-$\delta^{18}$O relationship. We changed the sentence as follows: "Figure 10 shows that simulation of the $\delta$D and $\delta^{18}$O relationship is generally correct in IsoGSM since $\delta$D and $\delta^{18}$O exhibit for all simulations/assimilation experiments the expected linear relationship."**

- *Fig 11: why is the noDA simulation so difference from the Nature run? What is the difference between the noDA and the Nature run, except for different initial conditions? I can understand different synoptic variations, but why such different mean $\delta$D-$\delta^{18}$O relationships?*
  **The noDA run is an ensemble simulation, consisting of 96 ensemble members, without any data assimilation while the Nature run is our constructed "truth", an free running IsoGSM simulation.**

- *l 365: this sentence is not grammatically correct.*
  **The sentence has been rewritten as follows: "With respect to the performance of the assimilation experiments, we find that the differences between the assimilation experiments and the Nature run are increasing as wider the value range of the d-excess spans."**

- *When $\delta D$ is assimilated, what happens with $\delta^{18}O$? Is it left free? Or is it assimilated assuming some $\delta D$-$\delta^{18}O$ relationship? What is the impact of the way $\delta^{18}O$ is assimilated (or not) on the results regarding $\delta D$-$\delta^{18}O$ relationships and d-excess?*
  **Only $\delta$D is assimilated, but since $\delta^{18}$O is included in the state vector (see description S2 in Toride et al. (2021)) $\delta^{18}$O is also influenced by the assimilation via the**

**covariance between $\delta$D and $\delta^{18}$O. In all our assimilation experiments we derive an improvement for both, $\delta$D and $\delta^{18}$O. Thus, we can also expect that the $\delta$D-$\delta^{18}$O relationship is to some extent improved, but this improvement will mostly be linear (based on kinetic fractionation) and thus there will be still uncertainties concerning the non-kinetic processes (and thus the d-excess). However, we do not expect that this does have any significant influence on the results derived in our study.**

- *l 408: wrong, moistening by rain evaporation is known to increase d-excess in the vapor, e.g. [Tremoy et al., 2014]*
  **We agree, it is not as easy as we stated. Thus, we changed the sentence as follows: "D-excess can act as fingerprints of earlier processes as e.g. cloud condensation, evaporation, mixing and air mass transport (Noone, 2012; Tremoy et al., 2014; Aemisegger et al., 2015; Salmon et al. 2019)."**

- *l 445: upward or downward? heating or cooling? If both, this sentence should just say "all regions"*
  **What we meant here is that high RMSDs are found within the regions of downward motion (that correspond to cooling regions) and within the regions of upward motion (that correspond to heating regions). These regions of high RMSD are not found throughout the entire troposphere and not throughout all longitudes, thus to write here "all regions" would not be correct. For $\delta$D these regions are found in the mid to upper troposphere and for $Q_2$ in the lower to mid troposphere. We changed the sentences as follows: "We found that these regions of high RMSD in $\delta$D are located in the mid to upper troposphere at around 150 W, 50 W and 150 E and coincide with regions of both, upward motion (diabatic heating) and downward motion (diabatic cooling) while the regions with high RMSD in $Q_2$ are located in the lower to mid troposphere between around 150 E to 180 W and coincide solely with regions of upward motion and diabatic heating, respectively."**

References:
Fueglistaler, S., Legras, B., Beljaars, A., Mocrette, J.-J., Simmons, A., Tompkins, A. M. and Uppala, S.: The diabtaic heat budget of the upper troposphere and lower/mid stratosphere in ECMWF reanalyses, Quarterly Journal of the Royal Meteorological Society, 135, 21-37, 2009.

Galewsky, J., Steen-Larsen, H. C., Field, R. D., Worden, J., Risi, C., and Schneider, M. (2016). Stable isotopes in atmospheric water vapor and applications to the hydrologic cycle. Reviews of Geophysics, 54(4):809–865.

Dee, S. G., Nusbaumer, J., Bailey, A., Russell, J. M., Lee, J.-E., and Konecky, B.: Tracking the strength of the Walker circulation with stable isotopes in water vapor, Journal of Geophysical Research, 123, 7254–7270, https://doi.org/10.1029/2017JD027915, 2018.

Noone, D. (2012). Pairing measurements of the water vapor isotope ratio with humidity to deduce atmospheric moistening and dehydration in the tropical mid-troposphere. Journal of Climate, 25(13):4476–4494.

Thorncroft, C. D., Nguyen, H., Zhang, C., and Peyrillé, P. (2011). Annual cycle of the west african monsoon: regional circulations and associated water vapour transport. Quarterly Journal

of the Royal Meteorological Society, 137(654):129–147.

Toride, K., Yoshimura, K., Tada, M., Diekmann, C., Ertl., B., Khosrawi, F., and Schneider, M. (2021): Potential of mid-tropospheric water vapor isotopes to improve large-scale circulation and weather predictability, Geophysical Research Letters, 48, e2020GL091 698, https://doi.org/10.1029/2020GL091698.

Tremoy, G., Vimeux, F., Soumana, S., Souley, I., Risi, C., Cattani, O., Favreau, G., and Oi, M. (2014). Clustering mesoscale convective systems with laser-based water vapor delta18O monitoring in Niamey (Niger). J. Geophys. Res., 119(9):5079–5103, DOI: 10.1002/2013JD020968.

Wright, J. S. and Fueglistaler, S.: Large differences in reanalyses of diabatic heating in the tropical upper troposphere and lower stratosphere, Atmospheric Chemistry and Physics, 13, 9565 – 9576, https://doi.org/10.5194/acp-13-9565-2013, 2013.

[Figure]

Figure 1: Improvement/degradation in skill in percent for each parameter in the troposphere (up to the 100 hPa level) derived from averaging the vertical skill profiles for the simulation runs with assimilation of the mocked IASI data compared to the simulation without any data assimilation (noDAvsDA experiment). Shown are the result for the latitude band 20°S to 20°N (top), 0° to 20°N (middle), 20°S to 0° (bottom).

[Figure]

Figure 2: Improvement/degradation in skill in percent separated by regions for each parameter in the troposphere (up to the 100 hPa level) derived from averaging the vertical skill profiles for the simulation runs with assimilation of the mocked IASI data compared to the simulation without any data assimilation (noDAvsDA experiment). Shown are the result for the latitude range 20°S to 20°N (top), 0° to 20°N (middle), 20°S to 0° (bottom).

[Figure]

Figure 3: Cross sections for heat source ($Q_1$), moisture sink ($Q_2$) and vertical velocity ($\omega$) derived from the Nature run for the latitude bands 20°S to 20°N (top), 0° to 20°N (middle), 20°S to 0° (bottom)

---

## Author Comment (AC2)

**Reply to Referee 2 Comments**

**Manuscript-No: wcd-2019-49**

**Can the assimilation of water isotopologue observation improve the quality of tropical diabatic heating and precipitation?**

**We thank referee 2 for the constructive, helpful criticism and the suggestion for revision. We have thoroughly revised the manuscript based on the comments given by the referees. A detailed point-by-point response to the comments by referee 2 are given below.**

*The aim of this paper is to show the benefit of stable water isotope observation assimilation for improving the representation of diabatic heating and precipitation in the tropics. A theoretical approach is chosen based on Observation System Simulation Experiments (OSSEs). The OSSEs are nearly the same as the ones presented earlier this year in Toride et al. 2021. While I do think that water isotopes contain valuable additional information on atmospheric circulation characteristics and moist diabatic processes in the atmosphere, I am very skeptical about their direct usefulness in data assimilation. In my view, there is no evidence provided in this paper that would support such a conclusion.*

**With the upcoming next generation of Metop satellites, isotope measurements from IASI will be available for the next decades and thus isotopes can definitely be valuable in data assimilation, especially together with water vapour. Since the highest improvements for the assimilation experiments were derived when both, isotopes and water vapour, are assimilated as was shown by Toride et al. (2021), it would be of course optimal if both species would be assimilated together. Note, IASI data are currently not operationally assimilated. However, the major intention of our study is not to show with which data set the highest improvement can be derived. Our intention is to understand the direct impact the assimilation of isotopes has on the meteorological analyses. This is why we only use the experiment from Toride et al. (2021) where only $\delta$D additionally to conventional observations is assimilated and compare this to an experiment that has been performed in the frame of this study where $\delta$D is assimilated without any other data. We hope that the revisions we made on the manuscript based on the referee's comments make the intention of our study and the outcome now more clear.**

*The major reasons, why I think that the paper is difficult to understand in the current form are:*

*1) Contradiction in stated hypothesis of the physical reason for the added value of isotopes in data assimilation and the outcome of the second OSSE experiment*
*As stated by the authors in the introduction, the rationale for the use of isotope observations to improve various meteorological fields such as T,q,u,v is that they are tracers of moist diabatic processes in the atmosphere. Thus, via improvements in diabatic heating rates in models, isotope assimilation leads to improvements in other fields. However, that is not what the authors observe in their second OSSE, in which they only assimilate $\delta$D. In the noDavsDa experiment the authors find an improvement in all variables except those ($\omega$, Q1, Q2), for which we would expect a direct physical link with the mid tropospheric $\delta$D distribution to exist. This contradiction is very disturbing for the readers and unfortunately not addressed at all by the authors. Based on this result, what do the authors think, is the reason for the improvements observed in the other meteorological fields?*

The time series in Fig. 9 and 10 actually show that the assimilation of $\delta$D is not failing. On a qualitative basis the assimilation of $\delta$D improves these parameters, too. Only when the performance is assessed quantitatively using the skill we derive a less good agreement for $\omega$, $\mathbf{Q}_1$ and $\mathbf{Q}_2$ than for the other parameters. A possible explanation is that the other meteorological parameters do not have as strong fluctuations as $\omega$, $\mathbf{Q}_1$ and $\mathbf{Q}_2$, especially since these ones variate around zero. Thus, an accurate simulation of $\omega$, $\mathbf{Q}_1$ and $\mathbf{Q}_2$ is much more difficult and therefore we derive only an improvement when already the underlying physics (dynamics) are correctly simulated. We make this point clearer now in the manuscript and changed/added the following text in the conclusion: "The noDAvsDA experiment shows that on a quantitative basis the assimilation of IASI $\delta$D alone cannot significantly improve the heating rates. However, the assimilation of $\delta$D has a positive effect on all other parameters including precipitation. Further, that we derive a qualitative agreement for $\omega$, $\mathbf{Q}_1$ and $\mathbf{Q}_2$ when IASI $\delta$D alone is assimilated may explain why nevertheless precipitation rates can be improved. Furthermore, together with the conventional observations from PREPBUFR an additional improvement for all parameters, including the heating rates, can be achieved and shows the benefit of the IASI $\delta$D data. This indicates that the correct simulation of the underlying physics is important for improving diabatic heating and vertical motion."

*2) Observation density*
*Since $\delta$D assimilation can only lead to substantial improvements in diabatic heating when assimilated together with conventional observations, the question about the observation density arises. This should be discussed and an assessment of the observation density differences in the PREBUFR experiments should be provided. I know that this is done in the supplement of Toride et al. 2021, but I think this is so essential that it cannot just be left out of the discussion in this paper. Increasing the number of conventional observations at the locations of assimilated IASI $\delta$D (e.g. q profiles from IASI) instead of $\delta$D would maybe lead to even larger improvements.*
We agree that this point should not be left out of the discussion and it is correct that with the assimilation of IASI q or both q and $\delta$D higher improvements can be derived. However, there is no point in repeating exactly the same what is already done in Toride et al. (2021), especially since referee 1 already thinks we are too close. We focus here solely on the assimilation of isotopologues since the intention of our study is to investigate the direct impact the assimilation of isotopologues have on the diabatic heating rates. Therefore, we use the experiment of Toride et al (2021) assimilating IASI $\delta$D additionally to conventional observations and compare this to an experiment performed in the frame of this study where we assimilate only $\delta$D without any other data. Nevertheless, to make clear that the assimilation of q alone or of both, $\delta$D and q, is more successful in terms of improvement we added in Sect. 3.2 the following text: "Note, that in terms of improvement, however, the assimilation of IASI $H_2O$ or even both, IASI $H_2O$ and IASI $\delta$D is more efficient and leads to higher improvements (Toride et al., 2021)". Further, to make the intention of our study more clear and to better describe the differences between the study by Toride et al (2021) and our study, we added the following paragraph in the introduction: "Here, we build on the study by Toride et al. (2021) and investigate this latter issue further, namely which information is hold by isotopologues? Especially, we are interested in answering the following question: Can the information stored in water isotopologues help to improve diabatic heating rates and/or precipitation rates? For that we use the assimilation experiment assimilating isotopologues from the study of Toride

et al. (2021) and compare this to an additional OSSE performed in the frame of this study where we assess the direct impact of the IASI isotoplogues on the meteorological variables. In the additional OSSE the IASI isotopologues are assimilated alone (without any conventional observations) and then compared to an ensemble simulation where no observations at all are assimilated."

*3) Motivation for chosen tropical region delimitation I missed a clear motivation for the chosen tropical regions, over which the $\delta D$ induced improvements in data assimilation are quantified. Why not focusing on known ascent dominated regions along the ITCZ vs. subsidence dominated regions further away from the equator? In the current form I did not gain any process-based insight from the regional categorization.*

We apologize that we have not been clear and thus caused some confusion. In our study we focus on the Walker circulation, thus on the circulation cells in east-west direction (longitudinal direction) and not the ones in north-south direction (latitudinal). The cross sections we show in the manuscript are longitudinal ones. To make this more clear now throughout the manuscript we added the suffix "longitudinal" before cross sections and added a paragraph in the introduction motivating our regions of choice and introducing the Walker circulation. We have added the following text: "In this study, we focus on the inner tropics (10°S to 10° N), to assess the impact of isotopologues on the assimilation in the region where diabatic heating is strong and where the Walker circulation is found. The Walker circulation is a longitudinal (east-west) circulation pattern consisting of several circulation cells spanning over the entire tropics. Convection and heavy precipitation associated with the rising branches of the Walker Circulation occur over Indonesia and the western Pacific, northern South America, and eastern Africa while sinking air and desert conditions prevail over the eastern equatorial Pacific and west Africa (Peixoto and Oort, 1992; Lau and Yang, 2003; Webster and Chang, 1988)". Additional to separating the tropics into the three regions over land (Asia, America and Africa), we also separate the tropics by upward and downward branches of the Walker circulations (see Discussion and Fig. 13 and 14 (now Fig. 12 and 13) in the manuscript).

*4) Missing discussion on precipitation improvements Even though improvements in modelled precipitation seem to be expected through improvements in diabatic heating profiles, I find the discussion about precipitation too sparse to allow for such a prominent place in the title.*

We improved our discussion of precipitation throughout the manuscript, especially in the discussion and conclusion. Nevertheless, we decided to remove "precipitation" from the title since we still do not discuss precipitation to that extent as we discuss diabatic heating.

*Minor comments:*

- *Many parts of the paper are a bit lengthy in writing and in the shown Figures. For example:*

  - *A lot of information is given about IASI, even though no real IASI data is used.*
    This is correct and the respective section has been omitted and the for this study required information on IASI has been moved to section 2.3 (now 2.2)

  - *I cannot see the differences in the profiles shown in Fig. 6.*
    This is correct, differences between the assimilation experiments and the Nature are, as for the tropics, quite low and become only visible when the

MD, RMSD and skill are considered. However, we use this figure to describe the different characteristics of the three regions considered and showed the three assimilation experiment for the sake of completeness. The text has been changed as follows to make this clear: "Figure 6 shows the averaged ensemble mean profiles for $Q_1$, $Q_2$ and vertical velocity averaged over the respective regions for August 2016. As for the tropics, differences in the averaged mean profiles between the Nature and assimilation runs are quite low and become only visible when the mean differences between the assimilation run and Nature run are considered (Fig. S5)." Further, based on the comments by referee 1 the description of the differences of the regions has been shortened.

– *What can I learn from Figures 9 and 10?*
Based on the comment by referee 1 we now show instead of the time series of the mean differences the time series of the absolute values (which we before had in the supplement). In the time series of the absolute values one can clearly see the positive effect the isotope assimilation has. Without any data assimilation the ensemble mean depicts only the climatological conditions. When the isotopes are added a significant improvement between the Nature and the assimilation experiment is found, especially for America. Due to the assimilation of $\delta$D the synoptic-scale variations are introduced correctly, but differences concerning the daily variations remain which lead to less improvement in the skill.

– *The role of Section 3.4 about the $\delta D$-$\delta^{18}O$ relation and dexcess is not clear to me and does not fit well into the storyline.*
We use the $\delta$D-$\delta^{18}$O and the d-excess to assess the performance of the assimilation experiments and to investigate the differences we find concerning the performance for the three regions considered in this study. We revised the section to make this point clearer and added the following text in the introduction (where we give an outline of the paper structure) to make our intention with this analyses clearer: "Finally, we exploit the $\delta$D-$\delta^{18}$O relationship (Daansgard et al., 1964) and d-excess (Craig et al., 1961) which serves on one hand as a further assessment and on the other hand helps us to better understand the differences in performance for the specific tropical longitude regions considered in this study."

• *I did not understand the difference between the individual ensemble members. Were they just initialized at different times from the nature run? If yes, why are they different from the nature run, then? Or are the initial conditions perturbed with respect to the nature run?*
The isoGSM simulation that has been used to generate the Nature has been performed for 2-years starting on 1 June 2015. The 96 ensemble members are initialised with the conditions from 1 June 2016 onwards (consecutively every 6 h), thus with the meteorological conditions prevailing one year later. These initial conditions can be considered as being independent from the Nature, but representing similar climatological conditions.